# No Need to Train Your RDB Foundation Model

**Linjie Xu** [1]  **Yanlin Zhang** [1]  **Quan Gan** [1]  **Minjie Wang** [1]  **David Wipf** [1]

## Abstract

Relational databases (RDBs) contain vast amounts of heterogeneous tabular information that can be exploited for predictive modeling purposes. But since the space of potential targets is vast across enterprise settings, how can we *avoid retraining* a new model each time we wish to predict a new quantity of interest? Foundation models based on in-context learning (ICL) offer a convenient option, but so far are largely restricted to single-table operability. In generalizing to multiple interrelated tables, it is essential to compress variably-sized RDB neighborhoods into fixed-length ICL samples for consumption by the decoder. However, the details here are critical: unlike existing supervised learning RDB pipelines, we provide theoretical and empirical evidence that ICL-specific compression should be constrained *within* high-dimensional RDB columns where all entities share units and roles, not *across* columns where the relevance of heterogeneous data types cannot be determined without extensive label information. Conditioned on this restriction, we then demonstrate that encoder expressiveness is actually not compromised by excluding trainable parameters. Hence we arrive at a principled family of RDB encoders that can be seamlessly paired with already-existing single-table ICL foundation models, whereby no training or fine-tuning is required. From a practical standpoint, we develop scalable SQL primitives to implement the encoder stage, resulting in the easy-to-use open-source *RDBLearn* foundation model capable of robust performance on unseen datasets out of the box.

## 1. Introduction

Foundation models for tabular data, capable of handling new predictive tasks *without retraining*, are increasingly prevalent (Hollmann et al., 2025; Jingang et al., 2025; Zhang

[1]University of Hong Kong, Shanghai X-Lab. Correspondence to: David Wipf <davidwipf@gmail.com>.

*Proceedings of the 43rd International Conference on Machine Learning*, Seoul, South Korea. PMLR 306, 2026. Copyright 2026 by the author(s).

et al., 2025b;c). When predicated upon some form of in-context learning (ICL), these models push labeled instances from a previously unseen dataset through a single forward pass of a pre-trained Transformer architecture, and then output predictions at one or more user-specified testing points. Despite their promising performance thus far, these existing *single-table* foundation models do not address wide-ranging enterprise relational databases (RDBs) involving multiple inter-connected tables (Garcia-Molina et al., 2009). For example, on e-commerce platforms candidate RDB prediction targets may cover future product purchases (Ni et al., 2019), customer retention (Dave et al., 2014), click-through rates (mjkistler et al., 2016; Zykov et al., 2022), user churn (Ni et al., 2019), or charge/pre-payment attributes (Motl & Schulte, 2015). As reliance upon such capabilities continues to grow, moving beyond single-table solutions represents an important yet under-served frontier facing the ML community (Gan et al., 2024); see Appendix A for further details.

To accommodate such RDB use cases, intuition suggests that a rich parameterized encoder is paramount, converting variably-sized RDB neighborhoods (possibly expressed as subgraphs), into dense fixed-length embeddings that mirror single-table settings. Indeed, most existing RDB predictive models trained via *per-dataset supervision* operate more-or-less in this fashion (Dwivedi et al., 2025a; Robinson et al., 2024a; Wang et al., 2024), with a graph neural network (GNN) or Transformer-related architecture serving as the de facto encoder. These approaches have been shown to outperform alternatives based on combining parameter-free multi-table feature aggregation with trainable single-table prediction heads (Wang et al., 2024; Zhang et al., 2023b).

However, as we will argue both theoretically and empirically herein, what is natural in the supervised learning regime *need not necessarily transition to success in distinct ICL-based RDB foundation models*. The high-level rationale is that a dense encoder representation can interfere with the original column space of RDB tables, conflating units, roles, and levels of useful information. When a supervision signal is present this need not be problematic, since the encoder can simply learn to first prune away useless dimensions specific to a given dataset. But prior to seeing ICL samples within an RDB foundation model, a necessarily *fixed* encoder is incapable of adjudicating column roles, which may vary from dataset to dataset (or even task to task within a given RDB;

see Figure 1). Hence we advocate for RDB encoders that only compress vertically *within* high-dimensional columns (where shared units facilitate interpretable aggregation), not horizontally *across* columns, where relevance is largely indeterminate without sufficient label information. Importantly, conditioned on this restriction to vertical compression, we formalize how a *parameter-free* encoder results in a minimal loss of expressiveness. Hence we can directly pair this class of encoder with the most powerful existing single-table foundation models *with no training required*. This leads to our key contributions:

- We define a restricted class of RDB encoder that explicitly preserves column identities and interpretable information flow to a single-table ICL-prediction head. Within this class, we establish that encoder expressiveness is not compromised by excluding trainable parameters. This facilitates direct compatibility with the best existing single-table foundation models.

- We quantify how RDB encoders *outside* of the proposed class can provably increase estimation error and/or sample complexity when uninformative feature columns are present in a database.

- Using scalable SQL primitives to implement the encoder stage, we introduce *RDBLearn*, an easy-to-use open-source RDB foundation model capable of handling completely new datasets out of the box with no training or fine-tuning whatsoever. The performance exceeds existing alternatives, including a non-reproducible, closed-source industry model that has been trained with access to unknown real-world datasets. We also include ablations to support the conceptual underpinnings of the proposed pipeline.

We remark that parameter-free RDB encoding methods have been proposed in the past for engineering supervised predictive pipelines (Kanter & Veeramachaneni, 2015; Kramer et al., 2001; Zahradník et al., 2023). What differentiates our contribution is that we are not defaulting to such methods merely as a simplifying heuristic as in prior work. Instead, we are rigorously examining *why* a suitable family of these parameter-free encoders may actually be *preferred* when the goal is ICL over unseen RDBs, particularly those with task-dependent partitions of useful and useless columns.

## 2. RDB Predictive Modeling

In a canonical supervised learning setting, we are given training data $\mathcal{D} = \{\boldsymbol{X}, \boldsymbol{y}\} \equiv \{\boldsymbol{x}_{i:}, y_i\}_{i=1}^n$ with instance feature rows $\boldsymbol{x}_{i:} \in \mathcal{X}^d$ and corresponding instance labels $y_i \in \mathcal{Y}$ for all $i$. The goal is then to learn a parameterized model $f_\theta$ such that $f_\theta(\boldsymbol{x}_{\text{test}}) \approx y_{\text{test}}$ at any test point $\{\boldsymbol{x}_{\text{test}}, y_{\text{test}}\}$, where in practice $y_{\text{test}}$ is unknown. *RDB predictive modeling* generalizes the above via the inclusion of

an additional set of auxiliary data tables $\mathcal{T} = \{\boldsymbol{T}^k\}_{k=1}^K$, where $\boldsymbol{T}^k \in \mathfrak{T}^{n_k \times d_k}$ denotes the $k$-th table associated with a given entity type. Each table row corresponds with a single instance of that entity (e.g., an individual user), and the columns encode instance attributes (e.g., elements of a user profile). These attributes are generally heterogeneous in nature, often consisting of continuous or discrete numerical values, categorical fields, text fragments, or temporal information. Note that without loss of generality, and for notational convenience later, we also assert that $\boldsymbol{T}^K \equiv \boldsymbol{X}$.

To complete its specification and make use of these auxiliary tables for making predictions, an RDB also includes a set of relations $\mathcal{R} = \{\boldsymbol{F}^k, \boldsymbol{p}^k\}_{k=1}^K$. Here each $\boldsymbol{p}^k \in \mathcal{P}^{n_k \times 1}$ denotes a primary key (PK) column, with elements uniquely identifying the rows of $\boldsymbol{T}^k$. Meanwhile, each column $\boldsymbol{f}_{:j}^k$ of $\boldsymbol{F}^k$ represents a foreign key (FK), whose elements are all given by values in some fixed PK column it references. In this way, the domain of every FK $\boldsymbol{f}_{:j}^k$ is some $\boldsymbol{p}^{k'}$.

### 2.1. A Generic RDB Supervised Learning Pipeline

A generic RDB predictive modeling pipeline predicated on *supervised learning* can be executed via the following steps:

1. Convert RDB $\{\mathcal{T}, \mathcal{R}\}$ as defined above to a heterogeneous graph $\mathcal{G}$. There are multiple ways of doing so (Wang et al., 2024), but the most common involves treating each table $\boldsymbol{T}^k$ as a node type and each row $i$ within a given $\boldsymbol{T}^k$ as a node with features $\boldsymbol{t}_{i:}^k$. We then form directed edges using each FK column $\boldsymbol{f}_{:j}^k$ and PK column $\boldsymbol{p}^{k'}$ it points to within $\mathcal{R}$. This approach is widely adopted (Cvitkovic, 2020; Dwivedi et al., 2025a; Fey et al., 2023; Zhang et al., 2023a;b).

2. Based on $\mathcal{G}$, sample $H$-hop subgraphs or ego-networks $\mathcal{G}_H(\boldsymbol{x}_{i:})$ that are centered at each target row $\boldsymbol{x}_{i:}$ within $\boldsymbol{X}$. For temporal RDBs, sampling should exclude nodes with time-stamps later than $\boldsymbol{x}_{i:}$.

3. Independent of the original RDB or individual subgraph sizes, compute *fixed-length* embeddings $\boldsymbol{z}_{i:} = g_\phi[\mathcal{G}_H(\boldsymbol{x}_{i:})] \in \mathbb{R}^{d_z}$, where the *encoder* $g_\phi$ is some form of GNN or Transformer architecture.

4. Stack embeddings to form the revised $\mathcal{D} = \{\boldsymbol{Z}, \boldsymbol{y}\} \equiv \{\boldsymbol{z}_{i:}, y_i\}_{i=1}^n$ and train end-to-end by minimizing the supervised loss

$$\mathcal{L}^{\text{SL}}(\theta, \phi) = \sum_{i=1}^n -\log q_\theta\Big(y_i | g_\phi\big[\mathcal{G}_H(\boldsymbol{x}_{i:})\big]\Big) \quad (1)$$

over parameters $\phi$ from the encoder and $\theta$ from a suitable prediction head/decoder $q_\theta$.

5. At inference time, update $\mathcal{G}$ to reflect any new collected data, including new unlabeled test rows of $\boldsymbol{X} = \boldsymbol{T}^K$, form the new subgraph $\mathcal{G}_H(\boldsymbol{x}_{\text{test}})$, and then compute $q_\theta\big(y_{\text{test}} | g_\phi\big[\mathcal{G}_H(\boldsymbol{x}_{\text{test}})\big]\big)$ for making predictions of $y_{\text{test}}$.

## 2.2. RDB Foundation Models

Moving beyond the supervised learning setting of the previous section, the goal of *RDB foundation models* is to retain applicability across multiple RDBs, with minimal or no retraining required for each new predictive task. In addition to direct LLM-prompting approaches (Wydmuch et al., 2024), we discuss two notable possibilities that involve an explicit pre-training step over multiple RDBs.

**Schema-agnostic models.** Provided the encoder $g_\phi$ is designed to digest a broad spectrum of input RDB schema (across entity types and associated features) using a shared representational form, then it is possible to simultaneously train $q_\theta$ and/or $g_\phi$ over multiple real-world RDBs (Ranjan et al., 2025; Wang et al., 2025; Wu et al., 2025). The resulting model can, at least in principle, be applied to new unseen RDBs without retraining, or perhaps more realistically, with modest fine-tuning for any given task.

**ICL-based models.** Building on the growing development of models exploiting ICL for single-table data (Hollmann et al., 2025; Jingang et al., 2025; Zhang et al., 2025c), it is natural to consider extensions to multi-table RDBs using a graph encoder $g_\theta$ as adopted in previously-introduced supervised learning models (Fey et al., 2025). Expanding on step 4 from Section 2.1, the ICL multi-table training objective becomes

$$\mathcal{L}^{\text{ICL}}(\theta, \phi) = \mathbb{E}_{\mathcal{T}, \mathcal{R} \sim p(\mathcal{T}, \mathcal{R})} \Big[ - \log q_\theta \Big( y_{\text{test}} | z_{\text{test}}, \mathcal{D} \Big) \Big]$$
$$\text{with } \mathcal{D} = \{z_{i:}, y_i\}_{i=1}^n, \quad z = g_\phi[\mathcal{G}_H(x)]. \quad (2)$$

Unlike in (1), the revised decoder module $q_\theta$ now consumes a set of labeled ICL samples $\{z_{i:}, y_i\}_{i=1}^n$ as well as a test point $z_{\text{test}} = g_\phi[\mathcal{G}_H(x_{\text{test}})]$, and is charged with predicting $y_{\text{test}}$; see prior work for background justification of forming $q_\theta$ in this way for single tables (Hollmann et al., 2022; Nagler, 2023). All the requisite quantities are extracted from RDBs sampled during pre-training, which may be synthetically generated from some distribution $p(\mathcal{T}, \mathcal{R})$ and/or combined with available real-world RDBs. Synthetic generation has the advantage of unlimited volume, potentially as an extension of single-table synthetic-generation training pipelines already in common use (Hollmann et al., 2025; Jingang et al., 2025; Zhang et al., 2025c). In contrast, real-world RDB data with wide coverage is relatively difficult to collect, as most sources are private within enterprises.

At inference time, a *completely new* RDB $\{\mathcal{T}, \mathcal{R}\}$ is provided, including one or more unlabeled test query points $x_{\text{test}}$ within $X = T^K$. Critically, *no further per-dataset training occurs*; instead, $\mathcal{D}$ and $z_{\text{test}}$ are computed as in (2) using encoder $g_\phi$, and then we form our final estimator as $q_\theta(y_{\text{test}} | z_{\text{test}}, \mathcal{D})$. In practice, the latter amounts to pushing $z_{\text{test}}$ and the ICL samples within $\mathcal{D}$ through a single forward pass of a Transformer architecture used for instantiating $q_\theta$.

## 2.3. Limitations

Thus far schema-agnostic models (Ranjan et al., 2025; Wang et al., 2025; Wu et al., 2025) have only been narrowly applied within small sets of RDBs for which varying degrees of pre-training and/or fine-tuning were applied. Hence their applicability outside of this regime on fundamentally different RDB types remains uncertain. Meanwhile for pure ICL-based models predicated on synthetic generation during pre-training, no existing open-source frameworks are actually available for transparent evaluation. We only have the unpublished *closed-source* KumoRFM approach (Fey et al., 2025) with key details missing; see Appendix A.

## 3. A Foundation for RDB Foundation Models

Given the established track record of ICL-based *single-table* foundation models, we intend to push similar principles into the more complex *multi-table* regime. In this regard, we will adopt an ICL-based prediction head $q_\theta$, where ICL samples within a given dataset share a fixed dimension as in prior work. But we depart from existing foundation models in how we design our encoder $g_\phi$.

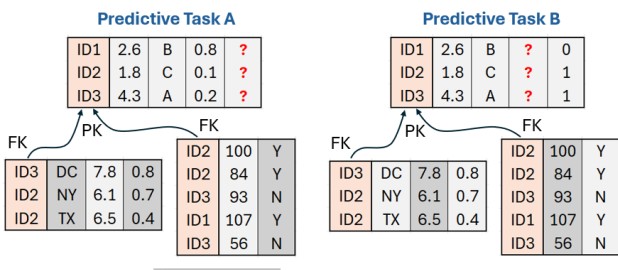

*Figure 1.* Example RDB (w/ $K = 3$ tables) where task-dependent column importance cannot be determined at the encoder stage. ICL samples are needed to resolve the intrinsic column ambiguity.

## 3.1. An RDB Encoder Specifically for ICL

Our guiding principle for RDB encoder design is as follows:

> There is no free lunch; RDB subgraphs may be large so any encoder **must** compress somewhere. Our insight is that at the encoder level, prior to seeing labeled ICL samples, we should only compress along the vertical dimension **within table columns**, where shared units facilitate interpretable aggregation guided by known PK-FK relations. Meanwhile, the encoder should **not** compress in the horizontal dimension **across table columns**, where roles, data types, and predictive relevance are broadly different and yet indeterminate without adequate per-dataset/task label info. See Figure 1 illustration.

We formalize these considerations through a particular encoder definition targeting ICL application.

**Definition 3.1.** We define a *JUICE* encoder function $g_{\text{juice}}$ (for "just use intra column encodings") if there exists a function $f_{\mathcal{R}}$, dependent on $\mathcal{R}$, such that the following hold:

1. $\boldsymbol{z}_{i:} \equiv \{z_{il}\}_{l=1}^{d_z} = g_{\text{juice}}[\mathcal{G}_H(\boldsymbol{x}_{i:})]$ for all rows of $\boldsymbol{X}$;

2. For each $l \in \{1, \ldots, d_z\}$ there is a $k \in \{1, \ldots, K\}$ and $j \in \{1, \ldots, d_k\}$ such that

$$\boldsymbol{z}_{:l} = f_{\mathcal{R}}\left(\boldsymbol{t}_{:j}^k\right) \quad \text{and} \quad \boldsymbol{z}_{:l} \perp\!\!\!\perp \mathcal{T} \backslash \boldsymbol{t}_{:j}^k. \tag{3}$$

Per this definition, if we convert each RDB subgraph to a fixed-length representation $\boldsymbol{z}_{i:}$ using JUICE, every column $\boldsymbol{z}_{:l}$ in the stacked representation matrix $\boldsymbol{Z}$ is a function of just *a single tabular data column from the original RDB*. Consequently, each $\boldsymbol{z}_{:l}$ reflects the identity and functional role of a single RDB column, and no other. Arbitration of which such columns are actually important to prediction accuracy is then deferred to $q_\theta$, where access to ICL samples and labels facilitates informed decision making.

**Caveats.** Before proceeding to JUICE implementation details, two notable qualifications are in order regarding the underlying motivation. Both of these address scenarios that might otherwise challenge strict reliance on Definition 3.1:

- There are other entry points for constructively accessing label information outside of an ICL-based prediction head. For example, limited historical labels exist within the subgraphs used as encoder input to certain RDB foundation models (Ranjan et al., 2025). However, unlike the full range of ICL samples, these *intra-subgraph* labels are restricted to localized RDB context (e.g., past instances of the same target entity) and need not be sufficient for differentiating uninformative features. We defer a rigorous treatment of this topic to future work.

- Herein we are implicitly assuming that external RDB descriptions and meta-data are unavailable for gauging column importance. This assumption facilitates operation even in ambiguous regimes without comprehensive annotations. But of course when such data are present, there is now a rich literature supporting the application of LLM agents as feature selectors/engineers (Han et al., 2024; Zhang et al., 2024), and reasonable featurization candidates may naturally deviate from Definition 3.1 (as an extreme illustrative example, consider an RDB task with attendant text explicitly stating that target labels are equal to an aggregation of the product of multiple columns from a joined table). Concurrent work even extends related ideas specifically to RDB predictive modeling (Kim et al., 2026).

### 3.2. A 1D GNN Implementation

We now address practical ways to actually construct JUICE embeddings. The high-level idea to first split a given $\mathcal{G}_H(\boldsymbol{x}_{i:})$ into separate 1D subgraphs associated with each tabular column dimension, and then apply a specialized GNN to these subgraphs independently.

**1D column-wise subgraph formation.** For each tabular data column $\boldsymbol{t}_{:j}^k$ we form the revised subgraph $\mathcal{G}_H^{k,j}(\boldsymbol{x}_{i:})$, which is equivalent to $\mathcal{G}_H(\boldsymbol{x}_{i:})$ but with truncated node features dependent only on $\boldsymbol{t}_{:j}^k$. Specifically, for nodes extracted from any arbitrary table $\boldsymbol{T}^{k'}$, the original $d_{k'}$-dimensional row features are simply reduced to a 1D feature via

$$\begin{aligned} \boldsymbol{t}_{i':}^{k'} &\to t_{i'j}^k \quad \forall i' \quad \text{if } k' = k \\ \boldsymbol{t}_{i':}^{k'} &\to 0 \quad \forall i' \quad \text{if } k' \neq k. \end{aligned} \tag{4}$$

By design, any encoder applied to the revised subgraph $\mathcal{G}_H^{k,j}(\boldsymbol{x})$ will only depend on column $\boldsymbol{t}_{:j}^k$, *retaining independence from all other RDB columns* as desired. We describe a general procedure for such encoder construction next.

**Meta-path GNN layers.** Although we have simplified our subgraphs via (4), each $\mathcal{G}_H^{k,j}(\boldsymbol{x}_{i:})$ nonetheless retains rich heterogeneous relationships through different PK-FK pairs within $\mathcal{R}$. These pairs determine a set of multi-relational meta-paths (Ferrini et al., 2024) between a target node defined by a row $\boldsymbol{x}_{i:}$, and the nodes associated with rows of other tables within a $H$-hop radius.[1] Analogous to tabular data columns with varying relevance, prior to seeing ICL samples the encoder cannot possibly know which meta-paths are most discriminative for a given predictive task.

A typical heterogeneous GNN (Busbridge et al., 2019; Hu et al., 2020; Schlichtkrull et al., 2018) would naturally interleave all of these meta-paths together in forming predictions. But this can be counter-productive outside of the supervised learning setting, where back-propagated gradients reflecting supervision labels are available to selectively discount the less important meta-paths on a dataset-by-dataset basis. Fortunately though, we can instead encode *separate* 1D representations associated with each meta-path using a meta-path GNN. Subsequently $q_\theta$ equipped with ICL samples can determine relevance, as candidate meta-paths remain conveniently column-aligned in the augmented feature space.

Given a length-$H$ meta-path $\rho_H$ extracted from $\mathcal{G}_H^{k,j}(\boldsymbol{x}_{i:})$, we initialize all 1D input-layer node embeddings denoted $\boldsymbol{\mu}^{(0)}$ using (4). From there, embedding updates (Ferrini et al., 2024) for a node $v$ on layer $h + 1$ are given by

$$\mu_v^{(h+1)} = \sigma\left(w_0^{(h)}\mu_v^{(h)} + \text{agg}\left[\left\{w^{(h)}\mu_u^{(h)}\right\}_{u \in \mathcal{N}_v^{(h)}}\right]\right), \tag{5}$$

where agg is a permutation invariant aggregation function (e.g., sum, mean) and $\sigma$ is an activation (e.g., linear, leaky-ReLU, etc.). Additionally, $\mathcal{N}_v^{(h)}$ denotes the set of neighbors

---

[1]A meta-path $\rho_H$ is a sequence of $H$ edges connecting properly-typed nodes in $\mathcal{G}_H^{k,j}(\boldsymbol{x}_{i:})$. For example, on an e-commerce platform we could have $user \xrightarrow{\text{purchased}} product \xrightarrow{\text{sold\_by}} seller$.

of node $v$ according to the relation or edge type associated with step $h$ along $\rho_H$ (Ferrini et al., 2024). We also assume meta-paths traverse each table at most once, before converting output layer embeddings to elements of $\boldsymbol{Z}$.

**Resulting JUICE encoder design.** Although not the most computationally efficient way to implement $g_{\text{juice}}$, we now summarize the constituent steps from a conceptual standpoint for transparency (full implementation details and complexity considerations will be addressed in Section 5):

1. Given a new RDB, select a row $\boldsymbol{x}_{i:}$ from $\boldsymbol{X} \equiv \boldsymbol{T}^K$ and initialize $\boldsymbol{z}_{i:}$ to $\boldsymbol{x}_{i:}$.

2. Extract $\mathcal{G}_H(\boldsymbol{x}_{i:})$ as in prior work on supervised RDB predictive models.

3. Choose a tabular data column $\boldsymbol{t}_{:j}^k$ from within an $H$-hop radius of $\boldsymbol{x}_{i:}$ over $\mathcal{G}_H(\boldsymbol{x}_{i:})$, and convert $\mathcal{G}_H(\boldsymbol{x}_{i:})$ to $\mathcal{G}_H^{k,j}(\boldsymbol{x}_{i:})$ using (4).

4. From within $\mathcal{G}_H^{k,j}(\boldsymbol{x}_{i:})$, select a meta-path $\rho_H$ between the node associated with row $\boldsymbol{x}_{i:}$ and the rows in table $k$, and propagate node embeddings using (5).

5. Collect the 1D node embeddings associated with $\boldsymbol{x}_{i:}$ from the output layer and concatenate to $\boldsymbol{z}_{i:}$.

6. Repeat the above, looping over meta-paths and data columns within $H$-hops of $\boldsymbol{x}_{i:}$.

7. Output final embedding $\boldsymbol{z}_{i:} = g_{\text{juice}}[\mathcal{G}_H(\boldsymbol{x}_{i:})]$.

If we compute JUICE embeddings for all rows of $\boldsymbol{X}$ as described above, the resulting $\boldsymbol{Z}$ will necessarily satisfy Definition 3.1. In fact, the interpretable column-wise information partitioning is even stronger: Each column of $\boldsymbol{Z}$ so-constructed is actually a function of a single auxiliary data column *and* a single meta-path connecting the target node with nodes within this data column. This organization dramatically simplifies the ICL stage, which need not disentangle useful and useless dimensions that have been nonlinearly coupled through a traditional dense encoder.

**Additional JUICE variations.** There exist other channels for enhancing JUICE expressiveness as well. Specifically, we may execute (5) with multiple different agg functions and concatenate the results to obtain a richer set of column-aligned representations. Depending on the task, different aggregations may be useful in sorting out various forms of homophily versus heterophily network effects (see Appendix D for further background context and related discussion). For example, for an RDB predictive task dominated by homophily relationships, mean or mode aggregation could potentially be quite valuable. Meanwhile, stdev, entropy, or quantile aggregations could be suitable for capturing a portion of network effects leaning in the heterophily direction. Concatenation of multiple such aggregation functions

has also been advocated in Corso et al. (2020), where it is shown that a single aggregator alone cannot differentiate various non-isomorphic graph structures. Moreover, it is reasonable to invoke LLM agents for proposing candidate sets of aggregation functions and/or associated meta-paths (albeit at a greater risk of leakage because of prior LLM exposure to benchmarking data). The SQL-based foundation of RDBLearn, discussed further in Section 5, is particularly amenable to this form of augmentation.

### 3.3. Encoder Training is Not Necessary

For a generic GNN or Transformer-based encoder with multi-dimensional node features, trainable linear filters are critical for increasing performance. However, by design our situation only requires 1D node features, which facilitates helpful simplifications per the following:

**Proposition 3.2.** *If $\sigma$ and* agg *are positively homogeneous functions, we initialize embeddings using (4), and $\{w_0^{(h)}, w^{(h)}\} \in \mathbb{R}_+ \ \forall h$, then without loss of generality (5) can be reparameterized with no internal weights.*

We remark that many of the most commonly used activations (e.g., ReLU, leaky-ReLU, linear) and aggregations (e.g., mean, sum, mode, min, max, stdev) are positively homogeneous, so this requirement is not a significant limitation. Additionally, while assuming positive internal weights does impose some modest form of constraint, sign information can be re-introduced by absorbing into revised $\sigma$ and agg definitions. Multiple such variations can be incorporated into JUICE to recover full expressiveness (in Section 3 we also advocate for inclusion of multiple aggregators).

As such, by virtue of Proposition 3.2, we can remove parameters from our 1D GNN implementation of JUICE without any appreciable loss of expressiveness. Note that even if such a reparameterization produces additional scale factors on the output layer (as opposed to internal weights), these can be absorbed into $q_\theta$. In fact, normalizing ICL feature dimensions (a common pre-processing tactic (Hollmann et al., 2025)) removes such scale factors anyway.

### 3.4. Combining JUICE with Single-Table ICL Model

By preserving column roles and identities (without conversion to traditional dense embeddings that fuse cross-column units and network effects), ICL samples $\{\boldsymbol{z}_{i:}, y_i\}_{i=1}^n$ produced by JUICE retain the canonical form for which single-table foundation models were originally designed. And critically, because this is achievable via a parameter-free encoding process, no additional training is required at any stage if we simply pair JUICE with one of these existing single-table models for instantiating the decoder $q_\theta$, e.g., TabPFN (Hollmann et al., 2025).

It is worth emphasizing that models like TabPFN are highly capable of processing multiple disparate tabular feature

columns, pruning away the unnecessary ones, and making predictions (Zhang et al., 2025a); they are not especially sensitive to the specific marginal distributions of each column that may change with vertical aggregations. As such, by construction JUICE largely preserves current TabPFN advantages and can leverage future enhancements as well.

# 4. Analytical Support for JUICE

Prior to seeing ICL samples with labels, it is not generally possible for any dataset-agnostic encoder (meaning one that has not undergone supervised training on a specific dataset) to consistently determine from individual subgraphs alone which constituent features are informative and which are not. The caveats mentioned in Section 3.1 notwithstanding, this is because the role and utility of any given feature itself, such as a column in an auxiliary table or a meta-path connecting different entity types, can vary from RDB to RDB, and even from task to task within a single RDB. In this section we further examine this limitation of fixed encoders in two complementary respects as related to the ICL setting.

## 4.1. Limitations of Cross-Column Dense Embeddings

We first consider a simplified data generation scenario introduced to isolate the consequences of using compressed embeddings from an arbitrary dense encoder. Similar consequences emerge from more complex setups as well at the cost of presentation clarity.

**Definition 4.1.** We define $\mathcal{D}(\pi)$ as a dataset generated using a function $\pi : \mathbb{R}^{\kappa} \to \mathbb{R}$ as follows. First draw samples $\boldsymbol{x}_{i:} \in \mathbb{R}^d$ from some $p(\boldsymbol{x}_{i:})$, where $i = 1, \ldots, n+1$ and $p(\boldsymbol{x})$ is assumed to be an absolutely continuous distribution satisfying $p_{\min} \leq p(\boldsymbol{x}) \leq p_{\max}$ for all $\boldsymbol{x} \in \mathcal{X}^{d}.$[2] We next select a set of $\kappa$ indices from $\{1, \ldots, d\}$ uniformly at random (without replacement) and form reduced features $\widetilde{\boldsymbol{x}}_{i:} \in \mathbb{R}^{\kappa}$. And finally, we compute labels $y_i = \pi(\widetilde{\boldsymbol{x}}_{i:})$ for all $i$. We then reserve $\{\boldsymbol{x}_{i:}, y_i\}_{i=1}^n$ for ICL while treating sample $n+1$ as $\{\boldsymbol{x}_{\text{test}}, y_{\text{test}}\}$.

By design of this generative process, uninformative features correspond with dimensions of $\boldsymbol{x}_{i:}$ that have been pruned in forming the reduced set $\widetilde{\boldsymbol{x}}_{i:}$, which alone is relevant to predicting $y_i$ within any given dataset. In the context of *supervised training* involving a single dataset draw, it is natural to learn compressed encoder representations $\boldsymbol{z}_{i:} = g_{\phi}(\boldsymbol{x}_{i:}) \in \mathbb{R}^r$ with $\kappa \leq r < d$. For example, an ideal scenario would have this encoder simply learning to discard all unnecessary dimensions of the original features $\boldsymbol{x}_{i:}$, such that making optimal predictions with the resulting compressed representation $\boldsymbol{z}_{i:} \approx \widetilde{\boldsymbol{x}}_{i:}$ is substantially easier. As a stark contrast, such simplifying encoder compression is *not* generally possible when we turn to *ICL settings*:

**Proposition 4.2.** *Assume* $\pi : \mathbb{R}^{\kappa} \to \mathbb{R}$ *is affine, with weights in general position. Then for any* $n > \kappa$, *there exists a fixed ICL decoder*[3] $f_{\theta}$ *such that*

$$\left| y_{\text{test}} - f_{\theta}\left(\{\boldsymbol{x}_{i:}, y_i\}_{i=1}^n, \boldsymbol{x}_{\text{test}}\right) \right| = 0 \qquad (6)$$

*with probability one over datasets* $\mathcal{D}(\pi)$ *generated according to Definition 4.1. Meanwhile, for any* $r < d$, *and any possible fixed encoder-decoder pair* $\{g_{\phi}, f_{\theta}'\}$, *we have*

$$\left| y_{\text{test}} - f_{\theta}'\left(\{\boldsymbol{z}_{i:}, y_i\}_{i=1}^n, \boldsymbol{z}_{\text{test}}\right) \right| > 0, \quad \boldsymbol{z} = g_{\phi}(\boldsymbol{x}) \in \mathbb{R}^r \quad (7)$$

*with probability at least* $1 - \binom{r}{k}/\binom{d}{k}$ *over the same data generative distribution.*

This result can be generalized to nonlinear data generation schemes and further elucidated with more precise error bounds. However, the core message of Proposition 4.2 is straightforward and can be conveyed without these extensions: *Ideal per-dataset compression of informative and uninformative feature columns is not possible with an encoder that must remain fixed across a distribution of input datasets, where column roles are subject to change.*

## 4.2. Sample Complexity Considerations

In Section 4.1 we assumed an encoder stage that formed dense, compressed representations with dimensionality $r < d$. We now turn to the case where $r = d$, i.e., no enforced compression across feature columns. In this regime we examine the degree to which a dense encoder $g_{\theta}$ may still negatively influence the sample complexity required to achieve a given estimation error.

At first glance this possibility may seem counter-intuitive. After all, during the training phase encoder-decoder pairs $\{g_{\phi}, f_{\theta}\}$ can (in principle) learn to coordinate in such a way that a parameterized $g_{\phi}$ producing dense cross-column representations is helpful. This is certainly true for new inference tasks that lie well *within* the distribution of the pre-training data. However, most real-world applications of ICL will inevitably deviate from this (likely synthetic) distribution, and it is here that a complex, parameterized encoder (as distinct from the decoder) can introduce problems.

To better understand this phenomena, we note that single-table ICL via a *decoder alone* can be relatively stable to distribution shifts (Zhang et al., 2025a), and asymptotically consistent under the right conditions (Nagler, 2023). In principle then, we can still obtain reasonable results even for new datasets that may substantially differ in distribution from a synthetic training set. Now consider the addition of an encoder, which sees *only a single input feature vector at a time*, not the full set of ICL samples; an OOD input here can

---

[2]Each such generated $\boldsymbol{x}_{i:}$ can be viewed as a set of features collected from one or more tables.

[3]For simplicity in this section, we adopt a deterministic decoder $f_{\theta}$, as opposed to the probabilistic version $q_{\theta}$ introduced earlier.

induce a substantially different encoder representation. And yet the encoder's behavior is only "known" to the decoder indirectly over the support of the pre-training data. And so from the decoder's standpoint, such a representation can lose any coordinated characteristics that might otherwise reduce estimation difficulty or sample complexity. In other words, in OOD regimes the encoder can essentially behave like an unknown transform, and if uninformative features exist within the original column-wise frame of reference, they will now be mixed by a process unknown to the decoder.

This difference can be dramatic. Working in the original feature frame leads to a sample complexity scaling with an exponential dependency on $\kappa$ (the number of informative features), while the OOD encoder can push this rate to be much worse, scaling exponentially with $d$ (the ambient dimension), even on the simplest of estimation problems. We quantify this phenomena as follows; see also Appendix B for empirical corroboration.

**Proposition 4.3.** *Let $\mathcal{F}$ denote the set of Lipschitz continuous functions on $[0,1]^\kappa$. Then there exists an ICL decoder $f_\theta$ such that (excluding smaller-order terms) we have*

$$\sup_{\pi \in \mathcal{F}} \mathbb{E}_{\mathcal{D}(\pi) \sim p}\Big[y_{test} - f_\theta\big(\{\boldsymbol{x}_{i:}, y_i\}_{i=1}^n, \boldsymbol{x}_{test}\big)\Big]^2 = O\left(n^{-2/\kappa}\right)$$

*where the data distribution $p$ is given by Definition 4.1 with $\mathcal{X} = [0,1]$. In contrast, with $\kappa = 1$ and $y = \widetilde{x}$, we have*

$$\inf_{f_\theta} \sup_{g_\phi \in \mathcal{B}} \mathbb{E}_{\mathcal{D}(\pi) \sim p}\Big[y_{test} - f_\theta\big(\{\boldsymbol{z}_{i:}, y_i\}_{i=1}^n, \boldsymbol{z}_{test}\big)\Big]^2 = \Theta\left(n^{-2/d}\right)$$

*where $\boldsymbol{z} = g_\phi(\boldsymbol{x})$ represents an encoder selected from the set of bi-Lipschitz bijections $\mathcal{B}$ mapping $[0,1]^d \to [0,1]^d$.*

## 5. RDBLearn: A Lightweight RDB Toolkit

We develop the open-source package *RDBLearn* to operationalize the merger of JUICE with single-table foundation models as advocated in Section 3.4. The result is an easy-to-use RDB toolbox with no training required. See Appendix E and companion documentation (Zhang et al., 2026) for RDBLearn usage and design specifics, including its optional agent-specific interface. We highlight several attributes here:

- *Intuitive interface*: The programming interface is designed to mirror the underlying formulation: core objects in the API correspond with concepts such as the relational database context $\{\mathcal{T}, \mathcal{R}\}$, ICL samples $\mathcal{D}$, and per-instance relational neighborhoods $\mathcal{G}_H(\boldsymbol{x})$. This makes it straightforward to relate experimental configurations and results back to modeling assumptions.

- *SQL-backed JUICE optimizations*: We implement JUICE principles using DFS primitives (Kanter & Veeramachaneni, 2015) through SQL execution over relational tables, leveraging the database engine for joins and aggregations.

System optimizations include: (i) Translating feature synthesis primitives into SQL queries with aggregation pushdown; (ii) Reusing intermediate results through caching or incremental materialization; and (iii) Compiling cutoff-time constraints into the SQL execution plan to avoid temporal leakage when predicting future targets.

- *ICL-model agnostic design*: Any single-table ICL predictor that follows the scikit-learn estimator interface can be directly incorporated. This supports controlled comparisons and upgrades under the same relational pipeline.

- *Temporal OOD reduction*: While base ICL predictors like TabPFN have been previously applied to forecasting tasks (Hoo et al., 2024), this requires introducing an array of cyclic temporal features to avoid OOD effects from unseen time stamps in the inference window. We sidestep this issue by encoding relative temporal differences rather than absolute times when forming ICL samples.

## 6. Testing with Real-World RDBs

For all experiments, RDBLearn extracts JUICE embeddings and converts to ICL samples using official benchmark training splits. We then choose $H \in \{2, 3\}$ and the RDBLearn base model from TabPFN-v2 (Hollmann et al., 2025), TabPFN-v2.5 (Grinsztajn et al., 2025), and LimiX (Zhang et al., 2025c) using dev sets, although RDBLearn is stable across these choices as shown below. *No other hyperparameters or tuning is involved whatsoever across all benchmarks*. Appendix F contains additional experiment details. Note that for simplicity, RDBLearn is currently implemented to completely ignore all text-based features; this is *not* the case for other baselines.

**Baselines.** We compare against the schema-agnostic **RT** (relational transformer) (Ranjan et al., 2025), **Griffin** (Wang et al., 2025), and **RelLLM** (Wu et al., 2025) models, all of which were pre-trained using the RelBench-v1 datasets (Robinson et al., 2024b). As for pure language model baselines, we report results from **LLM-A** (Team et al., 2025) (as tested in Ranjan et al. (2025)) and **LLM-B** (Wydmuch et al., 2024); both of these rely on serialized RDB neighborhood representations and/or ICL samples, and both can operate without any RDB pre-training or per-dataset fine-tuning on classification tasks. Although not a verifiable baseline, for reference we also include the closed-source **KumoRFM** industry model (Fey et al., 2025). And as a representative contrast outside of the foundation model scope of the others, we consider **RelGT** (Dwivedi et al., 2025a), a SOTA *fully-supervised* approach that shares the same dense encoder as KumoRFM. In this way, at a conceptual level the key distinction between KumoRFM and RDBLearn lies in the choice of encoder, with the latter alone based on JUICE foundations from Sections 3 and 4. All model hyperparameters were optimized per the specifications in prior work.

| | | no supervised training | | | | | | | |
|---|---|---|---|---|---|---|---|---|---|
| | | LLM-A | LLM-B | RelLLM | Griffin | RT | RandGNN | KumoRFM | RDBLearn | RelGT |
| rel-amazon | item-churn | 62.1 | 71.96 | 64.1 | 71.9 | 74.3 | 77.73 | 79.93 | 82.07 | 82.55 |
| | user-churn | 58.1 | 60.56 | 60.07 | 64.1 | 65.2 | 65.87 | 67.29 | 67.57 | 70.39 |
| rel-avito | user-clicks | 59.8 | 61.32 | 62.28 | 45.9 | 60.8 | 61.58 | 64.11 | 69.04 | 68.30 |
| | user-visits | 62.7 | 60.28 | 56.17 | 62.2 | 62.6 | 64.15 | 64.85 | 65.49 | 66.78 |
| rel-hm | user-churn | 59.8 | 64.34 | 55.95 | 60.4 | 63.1 | 66.54 | 67.71 | 68.05 | 69.27 |
| rel-stack | user-badge | 80.0 | 71.13 | 62.12 | 82.3 | 83.6 | 83.14 | 80.00 | 85.26 | 86.32 |
| | user-engage | 78.0 | 81.01 | 69.46 | 89.4 | 87.8 | 85.14 | 87.09 | 89.39 | 90.53 |
| rel-trial | study-outcome | 57.4 | 55.72 | 59.02 | 57.2 | 60.1 | 55.98 | 70.79 | 71.58 | 68.61 |
| | mean | 64.74 | 65.79 | 61.15 | 66.68 | 69.69 | 70.02 | 72.72 | 74.81 | 75.34 |

*Figure 2.* Entity classification results (AUC) on RelBench-v1; yellow is best, orange is second best among untrained models.

Lastly, we implement **RandGNN**, which amounts to a randomly-initialized GNN architecture for RDB feature encoding coupled with a single-table foundation model as used by RDBLearn; the specific GNN follows the architecture included with the RelBench-v2 release (Gu et al., 2026). Inclusion of this baseline allows us to disambiguate to what extent RDBLearn is merely capitalizing on strong single-table models versus the featurization principles behind JUICE. A substantial literature advocating for the efficacy of randomized-GNN layers specifically as viable feature encoders for relational prediction tasks provides further motivation (Bui et al., 2025; Degraeve et al., 2022; Xu et al., 2023). In the present context, such randomized GNN filters explicitly mix cross-column tabular information, violating the vertical-only design of JUICE.

**RelBench-v1 classification results.** Figure 2 presents results on the RelBench-v1 datasets, excluding rel-event and rel-f1, both of which have label leakage concerns noticed by ourselves and others (see Appendix F for details). Overall, RDBLearn outperforms the others, and is the only RDB foundation model that is competitive with the fully supervised RelGT approach. This is notable given that RDBLearn is only exposed to synthetic single-table data (during pretraining of the tabular base model); in contrast RelLLM, RT, and Griffin (and possibly others) have been exposed to each of the actual benchmarks as part of the pre-training adopted to produce the results in Figure 2. See Appendix A for further discussion of these models w.r.t. zero-shot learning. Lastly, we observe that RandGNN performs quite well (e.g., relative to RDB FMs from prior work) consistent with our analysis-based expectation that trained encoders may be less important in ICL settings (as opposed to supervised settings). And yet cross-column mixing is unhelpful, hence RDBLearn is still uniformly better than RandGNN.

**RelBench-v1 regression results.** Regression poses a unique set of challenges to RDB foundation models, and prior work often concedes that regression results are unsatisfactory without per-dataset fine-tuning, particularly for LLM-based approaches (Wu et al., 2025; Wydmuch et al., 2024). For this reason we have fewer baselines to compare against, which are further compromised by inconsistent met-

rics and reproducibility issues. Hence Figure 3 compares only against RandGNN (our implementation), KumoRFM, and RelGT for reference, as no other foundation models are directly comparable (see Appendix F for discussion of disanalogous Griffin and RT results). We remark that RDBLearn matches KumoRFM performance, and is even competitive with supervised RelGT, despite no regression-specific modifications or adjustments. It is unknown what regression allowances and/or regression-specific real-world pretraining data have been incorporated into KumoRFM.

| | | RandGNN | KumoRFM | RDBLearn | RelGT |
|---|---|---|---|---|---|
| rel-amazon | item-ltv | 66.130 | 55.254 | 48.559 | 48.922 |
| | user-ltv | 17.580 | 16.161 | 14.540 | 14.267 |
| rel-avito | ad-ctr | 0.038 | 0.035 | 0.034 | 0.035 |
| rel-hm | item-sales | 0.065 | 0.040 | 0.064 | 0.054 |
| rel-stack | post-votes | 0.071 | 0.065 | 0.068 | 0.065 |
| rel-trial | site-success | 0.434 | 0.417 | 0.424 | 0.326 |
| | study-adverse | 53.370 | 58.231 | 43.913 | 43.992 |
| | normalized mean | 1.218 | 1.088 | 1.074 | 1.000 |

*Figure 3.* Entity regression results (MAE) on RelBench-v1.

**RelBench-v2 results.** As requested by ICML reviewers we include preliminary results using the recently-released RelBench-v2 datasets (Gu et al., 2026). We address all classification (binary) and regression tasks with the following exceptions: we exclude rel-arxiv because it is primarily text-based and outside of our scope, and rel-mimic because it is not publicly-available. Results are presented in Figure 4, where RDBLearn is compared against Griffin and RandGNN, which we executed ourselves (with no additional training) via open-source code, and a fully-supervised GNN, the best performing model reported by Gu et al. (2026). Impressively, even with zero changes or tuning to accommodate this new testing domain, RDBLearn still achieves the best overall performance (even exceeding the supervised GNN on all but one task). We also emphasize that these RelBench-v2 tasks consist of hundreds of combined data columns (more than other benchmarks), many of which are likely uninformative.

**Additional 4DBInfer comparisons.** Prior work on RDB foundation models has focused on RelBench-v1 tasks. How-

| | Griffin | RandGNN | RDBLearn | GNN |
|---|---|---|---|---|
| **Classification (AUC)** | | | | |
| user-churn | 69.38 | 89.73 | 97.81 | 94.27 |
| beer-churn | 57.38 | 76.87 | 84.92 | 78.67 |
| brewer-dormant | 39.30 | 78.75 | 81.68 | 80.51 |
| **Regression (MAE)** | | | | |
| user-count | 13.01 | 13.94 | 7.231 | 7.374 |
| beer-ratings-total | 0.739 | 0.466 | 0.434 | 0.323 |

*Figure 4.* Results on RelBench-v2.

ever, to further establish the native versatility of RDBLearn, we apply our identical pipeline (again with zero modification whatsoever besides coupling with a suitable data loader) to the classification tasks drawn from the 4DBInfer benchmark (Wang et al., 2024). We compare against a suite of heterogeneous GNN and graph Transformer models specifically adapted for this benchmark with multiple graph extraction techniques; each such approach also benefits from per-dataset supervised learning and extensive hyperparameter optimization (see Appendix F). We also include RandGNN and Griffin as we did with RelBench-v2. Results are shown in Figure 5, where RDBLearn exhibits strong performance without training, leading to orders of magnitude greater efficiency (see Appendix F).

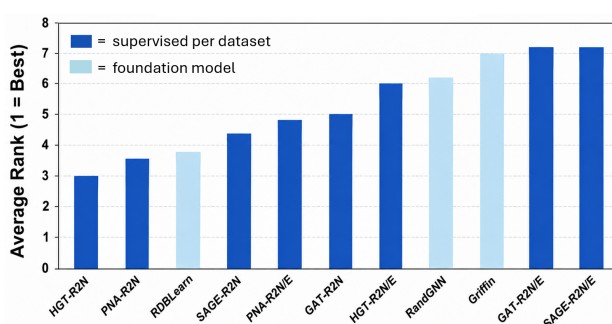

*Figure 5.* Entity classification ranking on 4DBInfer.

**JUICE is worth the squeeze.** If the principles underpinning JUICE (as used by RDBLearn) are sound, then we should expect a notable gap in performance when we contrast supervised learning versus ICL usage. Specifically, in the SL regime we should expect that an expressive parameterized $g_\phi$ may outperform $g_{\text{juice}}$, while the situation should largely flip when we turn to ICL cases. To empirically explore this phenomena, which has *not* been previously recognized, we consider the KumoRFM ICL model and the supervised RelGT model, both of which share the same $g_\phi$. Meanwhile, for $g_{\text{juice}}$ we can pair with both an ICL decoder (as within RDBLearn) and a strong supervised prediction head such as AutoGluon (Erickson et al., 2020). Given these four model instantiations, we plot the corresponding $g_{\text{juice}} - g_\phi$ performance differences split across SL and ICL cases in Figure 6. The outcome closely conforms with the expected performance inversion. In Appendix C we re-

peat an analogous experiment drawing on data from prior ICL-based graph foundation model studies; the outcome is similar. These results demonstrate the wider relevance of treating JUICE as an end goal, and not merely a simplifying approximation.

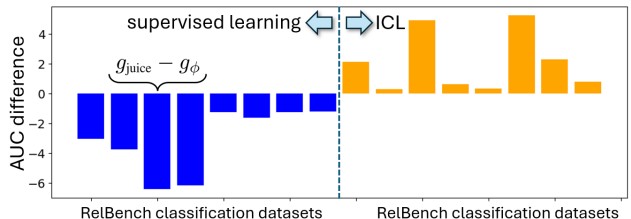

*Figure 6.* Performance inversion of JUICE embeddings.

**Ablation.** Figure 7 demonstrates the stability of RDBLearn as the base tabular prediction model and encoder hops $H$ are varied on the RelBench-v1 classification tasks. Notably, for every combination except one the performance exceeds all other RDB foundation models from Figure 2. These observations imply that, even without ever touching benchmark validation sets, nor adjusting any hyperparameters across different tasks, RDBLearn is capable of competitive test-set performance. And results will inevitably improve further still as new, more advanced single-table foundation models are released.

| | TabPFNv2 | TabPFNv2.5 | LimiX |
|---|---|---|---|
| **$H = 2$** | 73.94 | 72.03 | 73.03 |
| **$H = 3$** | 74.55 | 74.30 | 73.74 |

*Figure 7.* RDBLearn ablation (mean AUC on RelBench-v1).

## 7. Discussion

For RDB supervised learning, conventional wisdom points towards expressive parameterized encoders for converting variably-sized subgraph information into dense discriminative representations for end-to-end training. Meanwhile more traditional parameter-free RDB featurization steps are often viewed as outdated heuristics. However, we have argued that these designations need not generally hold, particularly when we move to ICL-based RDB foundation models. Our RDBLearn toolbox directly exploits these findings, relying only on scalable SQL-based featurization combined with frontier single-table architectures. Moving forward, both of these components are conducive to integration within agentic systems for either improving performance or addressing broader reasoning tasks involving RDBs (Li et al., 2025). In contrast, we contend that heavily-parameterized deep encoder representations (where feature roles become entangled and transparency is compromised) may pose greater challenges when agentic reasoning steps are involved.

## Impact Statement

This paper presents work whose goal is to advance the field of Machine Learning. There are many potential societal consequences of our work, none which we feel must be specifically highlighted here with one exception. A substantial portion of our contribution relates to increasing our understanding of RDB and tabular modeling components already being used in one way or another. While such understanding can in principle be exploited for nefarious purposes, overall we believe it to be a net positive.

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

# A. Extended Related Work

Although we have already cited key references needed to understand and contextualize our contributions in the main text, there are several additional points worth bolstering here along with supporting references.

**Growing relevance of RDB predictive modeling.** As of 2026, the majority of database management systems are relational,[4] and the information stored within contains countless possibilities for predictive modeling. Traditionally, developing such models was predicated on some form of propositionalization (Chepurko et al., 2020; Kanter & Veeramachaneni, 2015; Kramer et al., 2001; Kramer & Bessiere, 2020; Kumar et al., 2016; Liu et al., 2022), whereby fixed-length features are first extracted and aggregated within a single table such that standard tabular learning models can then be applied. The latter range from diverse boosting approaches (Chen & Guestrin, 2016; Ke et al., 2017; Prokhorenkova et al., 2018) to deep learning frameworks like DeepFM (Guo et al., 2017), SAINT (Somepalli et al., 2021), and FT-Transformers (Gorishniy et al., 2021).

More recently, with the advent of graph neural networks and wide-ranging graph Transformer models, there has been a notable shift towards end-to-end systems applied to graphs extracted from RDBs (Cvitkovic, 2020; Dwivedi et al., 2025a; Fey et al., 2023; Peleška & Šír, 2025; Wang et al., 2024; Zahradník et al., 2023; Zhang et al., 2023a). The underlying graph extraction process has also been the subject of ongoing exploration specifically for improving RDB prediction quality (Chen et al., 2025; Choi et al., 2025a; Gan et al., 2024). Of particular note, it has even been posited that the relevance of graph learning as a research domain in and of itself will fade unless focus is redirected towards (among other things) data originating in RDBs (Bechler-Speicher et al., 2025). For now though, evidence collected thus far suggests that in supervised settings, these recent end-to-end relational frameworks generally tend to outperform propositionalization, and conventional wisdom treats the latter as a bottleneck to be avoided where possible (Dwivedi et al., 2025b). Of course as we have shown both analytically and empirically, this need not still be the case when we turn to foundation models like RDBLearn formulated through ICL. See also Appendix C where we broaden the scope of these observations to graph foundation models.

**ICL and synthetic pre-training for RDB learning.** For pure ICL-based RDB foundation models based on synthetic generation during pre-training, there is presently only the KumoRFM approach (Fey et al., 2025). However, as a closed-source model there are few details available that might otherwise enable thorough assessment by the research community. For example, while pre-training was achieved using a mixture of synthetic and real-world RDB data, it is unclear to what extent the real-world portion maintains structural or task similarity with the limited evaluation benchmarks. This calls into question how generalizable KumoRFM actually is in practice. Nor is the synthetic generation pipeline used by KumoRFM available for scrutiny that might inform its potential for widespread efficacy. We remark that generating synthetic RDBs that reflect real-world properties is challenging, with unavoidable dependency on extra relational dimensions of variability not shared by successful single-table models (Gan et al., 2024).

**Zero-shot possibilities.** Mirroring ambiguity shared across the graph learning literature (Eremeev et al., 2025a; Xia & Huang, 2024; Xia et al., 2024), there does not appear to be a widely agreed upon definition of what constitutes true zero-shot RDB predictive modeling. Broadly speaking though, zero-shot learning refers to settings whereby the model must predict test samples involving classes that were not available during training. This is possible in situations where there exists suitable auxiliary information (e.g., textual attribute descriptions) that implicitly differentiate new classes.

In the context of RDBs specifically, the relational transformer (RT) model has been framed as possessing zero-shot capabilities, provided zero-shot relational learning is defined as "predicting new targets on a new RDB with a new schema, without weight updates" as proposed by Ranjan et al. (2025). Per this definition our RDBLearn framework would also technically qualify as zero-shot. But in such relational cases the auxiliary information relied upon for making predictions (by both RT and RDBLearn) includes exposure to entity labels with earlier time-stamps, e.g., from the entity we wish to classify and/or those extracted from prescribed neighborhood(s). This setup is conceptually a bit like label propagation on a temporal graph, which is not universally recognized as zero-shot per se, although admittedly it is reasonable to consider broader definitions as long as stipulations are clear.

Regardless of definitions, if RT pre-training is not explicitly conducted using each test RDB, the performance drops appreciably, even approaching a naive baseline whereby the historical entity mean serves as the prediction. For example,

---

[4] https://db-engines.com/en/ranking_categories

on RelBench-v1 classification tasks, this historical mean estimator achieves 66.7 average AUC, while comparable RT performance is 70.1 AUC; see Table 1 in Ranjan et al. (2025). In contrast, if we strictly enforce no target labels (past or present) available at inference time as a requirement for zero-shot relational learning, then the Griffin model (Wang et al., 2025) still technically qualifies, although performance without per-dataset fine-tuning is not competitive (64 average AUC under equivalent settings).

Finally, the RelLLM model (Wu et al., 2025) has also been described as a zero-shot learner when per-dataset fine-tuning is omitted. However, RelLLM is still pre-trained over the very same RelBench-v1 datasets upon which testing is conducted (even if the pre-training targets may vary). Moreover, without per-dataset fine-tuning, RelLLM falls behind even Griffin (63.2 average AUC). We reiterate though, outside of the definition proposed by Ranjan et al. (2025), our RDBLearn would generally not be considered a pure zero-shot method given its reliance on ICL samples. Even so, unlike these other approaches, RDBLearn does *not* depend on seeing the actual inference-time RDBs during pre-training to achieve SOTA performance. Rather, it is entirely based on a synthetic pre-training pipeline whereby there is no possible leakage or favorable bias introduced from inference-time RDBs.

## B. Empirical Corroboration of Dense Encoder Performance Degradation

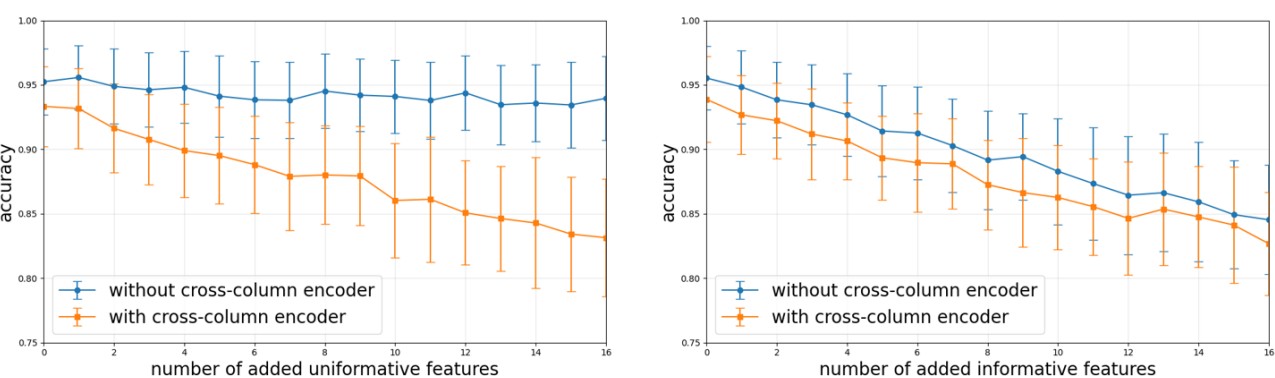

*Figure 8.* Impact of adding uninformative (*left*) versus informative (*right*) feature columns; results averaged over 100 trials.

To further explore our analytical findings from Section 4, we conducted the following simulation study involving two distinct testing scenarios. For the first, we initially generate ICL samples $\{\boldsymbol{X}, \boldsymbol{y}\} = \{\boldsymbol{x}_{i:}, y_i\}_{i=1}^n$, where elements of $\boldsymbol{X}$ are drawn iid from $\mathcal{N}(0, 1)$ and $y_i = \mathbb{I}\big[1/\big(1 + \exp[-\boldsymbol{w}^\top \boldsymbol{x}_{i:}]\big)\big]$ is a binary class label. The weight vector $\boldsymbol{w}$ is also drawn iid from $\mathcal{N}(0, 1)$. For all $i$ we then compute representations $\boldsymbol{z}_{i:} = g_\phi(\boldsymbol{x}_{i:})$ that mix cross-column information (unlike JUICE). This encoder is composed of a randomized linear filter, followed by a leaky-ReLU nonlinearity, and another random linear filter. By design this encoder representation is invertible so no information is lost.

We next compute the ICL-based prediction accuracy of a decoder $f_\theta$ (implemented here via TabPFNv2) at new test points $\{\boldsymbol{x}_{\text{test}}, y_{\text{test}}\}$ as additional uninformative columns are randomly inserted into $\boldsymbol{X}$ and the corresponding elements of $\boldsymbol{x}_{\text{test}}$. We repeat the same procedure using ICL samples $\{\boldsymbol{Z}, \boldsymbol{y}\}$ evaluated at corresponding test points $\{\boldsymbol{z}_{\text{test}}, y_{\text{test}}\}$. Figure 8(*left*) displays the results (averaged over 100 independently generated datasets for each plotted point) with and without the addition of encoder $g_\phi$. Of particular note, when no uninformative features are added (i.e., where the x-axis is zero on the left-hand plot), the encoder mixing has no appreciable effect; however, as more distracting columns are added, a clear negative trend emerges as expected per the analysis of Section 4.

Meanwhile, our second testing scenario operates as a form of control. Specifically, instead of adding uninformative features to $\boldsymbol{X}$ as before, we now introduce additional columns upon which a revised $\boldsymbol{y}$ computation now explicitly depends (this is accomplished by extending the length of $\boldsymbol{w}$ during generation). In this regime, the core prediction problem naturally becomes harder, since the decision function becomes increasingly complex and high-dimensional. But critically, the inclusion of the cross-column encoder now has little effect as shown in Figure 8(*right*). This is because the encoder is no longer mixing informative and non-informative features that would otherwise complicate the predictive task faced by $f_\theta$.

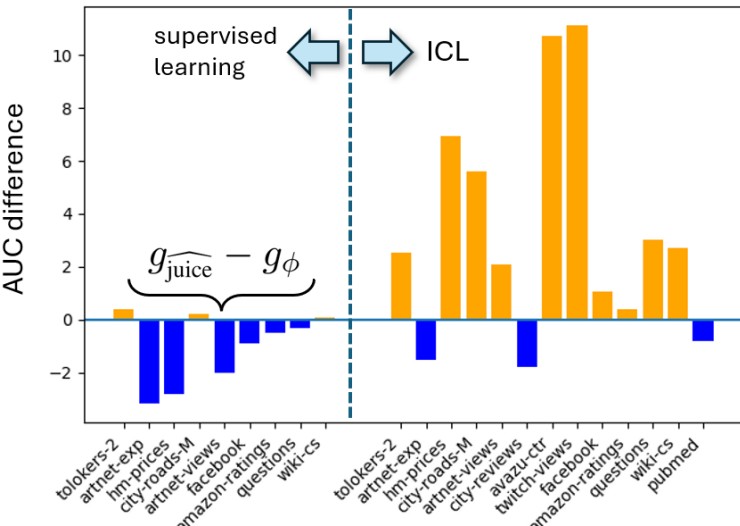

*Figure 9.* Performance inversion of JUICE-like embeddings on homogeneous graph datasets. Each bar plot is based on results extracted from Eremeev et al. (2025b) involving GraphPFN and G2T models formed with a LimiX ICL decoding architecture.

## C. Lessons from Graph Foundation Models Suggest that JUICE Is Worth the Squeeze

In Figure 6 from Section 6 we empirically demonstrated how the relative performance between a dense parameterized RDB encoder $g_\phi$ and our $g_{\text{juice}}$ dramatically shifts when moving from supervised learning to ICL settings. Ideally we would like to extend these results via testing over additional RDB foundation models beyond KumoRFM, upon which Figure 6 is based. However, because no other comparable frameworks exist, it is unfortunately not possible to do so without developing our own completely new synthetic generation and pre-training pipeline specific to RDBs. Of course it remains an open question how to actually implement these modules, and well beyond the scope of this paper (recall that KumoRFM is a closed-source industry model with an undisclosed pre-training process).

Fortunately though, there does exist work on *graph foundation models* that we can leverage to bootstrap an analogous comparison, allowing us to further establish that the SL-to-ICL performance inversion predicted by our analysis represents a wide-ranging phenomena. Although not a perfect surrogate, data from homogeneous attributed graphs can be viewed as isomorphic to simplified RDBs involving a single data table type.

To this end, we now introduce a novel re-combination of performance results from the GraphPFN model (Eremeev et al., 2025b), which relies on a dense parameterized encoder $g_\phi$ to compress neighborhood subgraph information, and the G2T approach (Eremeev et al., 2025a), which uses a parameter-free encoder with similarities to JUICE. We loosely refer to the latter as $g_{\widehat{\text{juice}}}$, as neighboring node features are aggregated in a column-wise fashion; likewise for additional structured features such as node degree and PageRank score.

What is particularly attractive about this setup is that we have access to performance results whereby the core ICL decoder is shared across all models. This restricts variation along the two dimensions we wish to probe, namely, the $g_{\widehat{\text{juice}}} - g_\phi$ performance (difference) under supervised learning, and $g_{\widehat{\text{juice}}} - g_\phi$ under ICL. This is possible because the GraphPFN model is pre-trained from a LimiX check-point (Zhang et al., 2025c), and results from Tables 2 and 3 in Eremeev et al. (2025b) cover both ICL and fine-tuning (i.e., per-dataset supervision) cases. Meanwhile, these same tables include a LimiX version of G2T under the same conditions. Hence we pull these results and compute the respective differences stratified over each dataset and learning type as shown in Figure 9 (the number of supervised learning datasets is fewer since some cases ran OOM).

Just as in Figure 6, which involves completely different datasets and models, Figure 9 reveals the same clear performance inversion predicted for JUICE-based embeddings. This consistency helps solidify our conclusions and widens the applicability of underlying JUICE design principles, suggesting that even in the realm of graph foundation models, dense parameterized encoders need not be necessary, at least outside of fine-tuning or supervised training regimes. We also remark that graph foundation models relying on parameter-free graph features served to single-table foundation models (like G2T) have been proposed in multiple concurrent works (Choi et al., 2025b; Eremeev et al., 2025a; Hayler et al., 2025). However, in all of

these cases the design is motivated by simplicity and guided by natural precursors from supervised graph learning (Frasca et al., 2020; Yoo et al., 2023). Unlike herein, prior work does *not* establish formal principles for grounding such approaches, nor explicit elucidation of the critical distinction between supervised and ICL usage thereof.

## D. Arbitrating Homophily vs Heterophily Relationships

Analogous to the feature- and meta-path-level ambiguities already discussed in the main text, there exists additional sources of uncertainty with respect to how network effects manifest in RDB predictive tasks. Depending on the task and dataset, unknown labels may be influenced by either heterophily or homophily relationships between neighboring nodes captured by distinct meta-paths. In the context of heterogeneous graphs defining RDB relations, homophily refers to the tendency of neighboring nodes of the same node type (not necessarily 1-hop neighbors) sharing the same label, while heterophily represents the converse. Differentiating such network effects is possible during supervised training on a per-dataset basis as gradients back-propagate through $q_\theta$ to $g_\phi$. In this way it is possible in principle for a single architecture, albeit with dataset-specific weights, to handle varying degrees of homophily on a case by case basis. But the ICL setting is completely different. When no per-dataset label-aware gradients flow to $g_\phi$, the encoder is blind to the importance of distinct meta-paths or the extent to which homophily vs heterophily dominates, and therefore is incapable of selectively preserving features that favor one effect over another. Hence suitable representations for *both* scenarios are needed prior to the ICL decoder. The JUICE philosophy suggests that these should be confined to distinct columns to the extent possible, such that subsequent ICL-based decoding is not unduly complex. Exploring this topic represents an interesting direction for future research.

## E. RDBLearn Framework Usage

The RDBLearn toolbox can be accessed at the following link:

https://github.com/HKUSHXLab/rdblearn

Alternatively, to use the affiliated RDBLearn skill, agents can fetch and follow instructions from:

https://github.com/HKUSHXLab/rdblearn/blob/main/SKILLS_INSTALL.md

Figure 10 illustrates the intended RDBLearn workflow. See also companion documentation (Zhang et al., 2026) for a more gentle introduction to RDBLearn application as well as comprehensive details regarding its implementation using DuckDB[5] as discussed in Raasveldt & Mühleisen (2019) and FeatureTools[6] as introduced by Kanter & Veeramachaneni (2015). In brief here, a user provides a task table $X$ (whose rows identify target records), associated labels $y$, and a broader relational database context. The estimator then automatically performs relational featurization following the spirit of JUICE and subsequently invokes a tabular ICL backend for prediction. Other particulars are as follows:

- *Relational database context*: The argument rdb corresponds with an RDB $\{\mathcal{T}, \mathcal{R}\}$, i.e., a collection of tables and their relations. Each instance $x$ (a row in $X$) is interpreted relative to this database context.

- *ICL samples*: The labeled training set $\{X, y\}$ supplied to fit provides the seed for internally computing in-context labeled examples $\mathcal{D} = \{Z, y\}$ used by the tabular ICL backend (see below for more details). At prediction time, the backend conditions on these examples to estimate $y_{\text{test}}$ for each query row in $X_{\text{test}}$.

- *Associating instances with relational context*: RDBLEARN exposes two complementary ways to associate each target instance with relevant relational records. Firstly, key_mappings provides *schema-based anchoring* by mapping columns in $X$ to primary keys in entity tables, thereby specifying which records are structurally reachable through relations in $\mathcal{R}$. Secondly, cutoff_time_column provides *time-conditioned context* by restricting the neighborhood to records whose timestamps are *strictly less than* the instance-associated cutoff time, a common requirement for leakage-free relational prediction in temporal settings. Together, these associations determine the relational neighborhood $\mathcal{G}_H(x)$ used by relational featurization.

- *Relational featurization and tabular prediction*: Given $\mathcal{G}_H(x)$, RDBLearn constructs a fixed-length representation $z$ internally. The downstream base_estimator then operates on the resulting tabular representation, and can be swapped as long as it follows the scikit-learn estimator interface.

---

[5]https://duckdb.org/
[6]https://github.com/alteryx/featuretools

```
from rdblearn.datasets import RDBDataset
from rdblearn.estimator import RDBLearnClassifier
from tabpfn import TabPFNClassifier

dataset = RDBDataset.from_relbench("rel-amazon")
task = dataset.tasks["user-churn"]

clf = RDBLearnClassifier(
    base_estimator=TabPFNClassifier())

target_col = task.metadata.target_col
X_train = task.train_df.drop(columns=[target_col])
y_train = task.train_df[target_col]

clf.fit(
    X=X_train,
    y=y_train,
    rdb=dataset.rdb,
    key_mappings=task.metadata.key_mappings,
    cutoff_time_column=task.metadata.time_col,
)

X_test = task.test_df.drop(columns=[target_col])
pred = clf.predict(X=X_test)
```

*Figure 10.* Minimal usage of RDBLEARN on a relational benchmark task.

## F. Experiment Details

### F.1. RDBLearn Setup

**Encoding settings.** For instantiating JUICE, we fix the activation function $\sigma$ to be linear and choose agg functions $\{\text{sum}, \text{mean}, \text{mode}, \text{min}, \text{max}, \text{std}\}$ for continuous-valued columns and $\{\text{counts}, \text{mode}\}$ for categorical columns. These common/intuitive selections remain unchanged across all benchmarks, datasets, and tasks reported herein. That being said, our preliminary testing (not shown) actually indicates that including additional aggregation functions can improve performance. However, we defer further exploration to future work.

We explicitly enable date/time features encoded as temporal differences to reduce OOD effects. However, we have chosen to simply disable all raw text, special text, and $n$-gram features. Interestingly, RDBLearn still achieves SOTA foundation model performance without these, indicating that textual attributions may not be central to making good predictions on current benchmarks (certainly this is often true of tabular data more broadly). In the future though, we can of course easily include a pre-trained language encoder to featurize text analogous to any other column type to boost prediction quality.

**ICL Decoder settings.** We evaluate three foundation model variants for the RDBLearn ICL-based decoding step:

1. **TabPFNv2** (checkpoint: tabpfn-v2-classification/regression-finetuned-zk73skhh);

2. **TabPFN v2.5** (checkpoint: tabpfn-v2.5-classification/regression-default);

3. **Limix** (checkpoint: LimiX-16M).

For consistency over all benchmarks we randomly down-sample training splits to 10k. Although some ICL base models can handle up to 50k samples, and this limit is regularly increasing, we found that 10k was sufficient for good performance.

**Equipment.** All experiments were conducted on a single NVIDIA 4080 GPU with 32GB memory.

### F.2. RelBench-v1 Testing

We follow the official temporal splits (train, dev, test) from Robinson et al. (2024b) to facilitate direct comparison with existing published work. Moreover, results presented herein represent tuned models as reported by original authors with one exception. Griffin model results in Figure 2 were obtained from Ranjan et al. (2025), where head-to-head alignment with the relational transformer was conducted under so-called zero-shot settings (see Appendix A for further discussion of zero-shot definitions). The Griffin paper itself (Wang et al., 2025) does not include this type of experimentation, focusing more on optimizing performance through per-dataset fine-tuning.

**Benchmark leakage issues.** As part of RelBench-v1 (Robinson et al., 2024b), the rel-event dataset has been found to have temporal leakage issues (Ranjan et al., 2025). There is also reason to believe that rel-f1 may be compromised as well, given that fine-tuned models are capable of essentially perfect accuracy (99.62 AUC) (Fey et al., 2025) despite these data being tied to sporting events (F1 racing) with non-negligible degrees of uncertainty over temporal splits. For these reasons, and following related studies elsewhere, we omit rel-event and rel-f1 from our empirical comparisons.

**Regression reproducibility.** Although promising RelBench-v1 regression results are reported in Ranjan et al. (2025) for a reduced set of models, the MAE metric is not used (e.g., as was used previously by KumoRFM). Moreover, for most datasets we were not able to reproduce the reported $R^2$ metric results even for the simple baseline "entity mean" estimator (and at the time of this writing, public code is not available for doing so).[7] The two exceptions are the rel-trial dataset tasks, namely, site-success and study-adverse. On these two datasets RDBLearn outperforms both Griffin and RT as shown in Figure 11.

|  | Griffin | RT | RDBLearn |
|---|---|---|---|
| **site-success** | 2.6 | 5.2 | 5.5 |
| **study-adverse** | -2.5 | 3.4 | 20.0 |

*Figure 11.* Regression results ($R^2$ scores, higher is better) on rel-trial tasks from RelBench-v1.

### F.3. 4DBInfer Testing

Following Wang et al. (2024), we adopt baselines formed from widely-used heterogeneous GNN architectures, including **R-SAGE** or **R-GCN** (Schlichtkrull et al., 2018), **R-GAT** (Busbridge et al., 2019), **HGT**, (Hu et al., 2020), and **R-PNA** (Corso et al., 2020). Each model type is then independently paired with each of two graph extraction techniques. The first is **R2N** (row-to-node), initially introduced by Cvitkovic (2020) and discussed in Section 2.1. The second is **R2N/E** (row-to-node/edge) as detailed in Gan et al. (2024). The importance of exploring multiple graphs is now well-established (Choi et al., 2025a). Collectively, this results in a total of 8 supervised baselines as listed in Figure 5, along with Griffin and RandGNN as representative foundation models executed by ourselves. Other more traditional baselines reported in Wang et al. (2024) have generally worse accuracy than these.

For specific datasets, we focus on 5 of the 4DBInfer classification tasks. These include click-through-rate prediction on Outbrain (mjkistler et al., 2016), user churn on Amazon Book Reviews (Ni et al., 2019), user churn and post popularity prediction on StackExchange,[8] and conversion prediction on RetailRocket (Zykov et al., 2022). For reference, the raw results across all baselines and tasks are shown in Table 1.

*Table 1.* Raw AUC values for 4DBInfer classification tasks used in producing Figure 5.

| Dataset | Task | Supervised | | | | | | | | Foundation Model | | |
|---|---|---|---|---|---|---|---|---|---|---|---|---|
|  |  | GAT(R2N) | GAT(R2N/E) | HGT(R2N) | HGT(R2N/E) | PNA(R2N) | PNA(R2N/E) | SAGE(R2N) | SAGE(R2N/E) | Griffin | RandGNN | RDBLearn |
| Amazon | Churn | 0.7622 | 0.7192 | 0.773 | 0.6864 | 0.7645 | 0.7157 | 0.7571 | 0.7314 | 0.5332 | 0.7742 | 0.7741 |
| Outbrain | CTR-100K | 0.6146 | 0.6308 | 0.626 | 0.6323 | 0.6249 | 0.6322 | 0.6239 | 0.6271 | 0.5222 | 0.4929 | 0.5447 |
| Retailrocket | CVR | 0.8284 | 0.7536 | 0.8495 | 0.8342 | 0.8367 | 0.8427 | 0.847 | 0.8091 | 0.3992 | 0.6621 | 0.8469 |
| StackExchange | post-upvote | 0.8853 | 0.6883 | 0.8817 | 0.6603 | 0.8896 | 0.7045 | 0.8861 | 0.6798 | 0.7166 | 0.6106 | 0.8845 |
|  | user-churn | 0.8645 | 0.8528 | 0.867 | 0.856 | 0.8664 | 0.8657 | 0.8558 | 0.8485 | 0.7218 | 0.7741 | 0.8796 |

**Timing considerations.** For many baselines, direct timing comparisons are elusive because of different computing environments and other confounds. That being said, we have confirmed that a single training run involving 4DBInfer baseline models described above operates on the scale of $10^3 - 10^4$ seconds. Combined with the 100-fold hyperparameter sweep (e.g., covering number of layers, hidden dimension, learning rate, etc.) needed to achieve reported results, the overall budget enters the $10^5 - 10^6$ second range. Meanwhile on comparable machines (a single NVIDIA 4080 GPU with 32GB memory) total RDBLearn latency with the same benchmarks is on the order of $10^2$ seconds. This latency can be improved with further optimizations, but doing so lies outside the scope of the present work.

---

[7]We remark that the entity mean represents a critical signal for the RT model as shown in ablations from Ranjan et al. (2025).

[8]https://data.stackexchange.com/

# G. Technical Proofs

**Proposition G.1.** *If $\sigma$ and agg are positively homogeneous functions, we initialize embeddings using (4), and $\{w_0^{(h)}, w^{(h)}\} \in \mathbb{R}_+ \ \forall h$, then without loss of generality (5) can be reparameterized with no internal weights.*

*Proof:* We begin by examining two special cases and assuming that $\sigma$ and agg are positively homogeneous of degree 1. First, if $\mu_v^{(h)} = 0$, then the node $v$ update from (5) is equivalent to

$$\mu_v^{(h+1)} = \sigma \left( \text{agg} \left[ \left\{ w^{(h)} \mu_u^{(h)} \right\}_{u \in \mathcal{N}_v^{(h)}} \right] \right) = w^{(h)} \sigma \left( \text{agg} \left[ \left\{ \mu_u^{(h)} \right\}_{u \in \mathcal{N}_v^{(h)}} \right] \right) \tag{8}$$

since both $\sigma$ and agg are positively homogeneous and $w^{(h)} \geq 0$ by assumption. Alternatively, if instead $\mathcal{N}_v^{(h)} = \emptyset$, meaning node $v$ has no neighbors at point $h$ on the meta-path, then

$$\mu_v^{(h+1)} = \sigma \left( w_0^{(h)} \mu_v^{(h)} \right) = w_0^{(h)} \sigma \left( \mu_v^{(h)} \right). \tag{9}$$

By definition of any $H$-hop meta-path and our corresponding meta-path GNN, we execute (5) $H$ times. Combined with the proposed initialization strategy given by (4) and assumption of no loops along meta-paths mentioned below (5), every propagation step that results reduces to either (8) or (9). To see this, note that at any step $h$ along a meta-path, by definition $\mathcal{N}_v^{(h)}$ only depends on a single active edge-type for all $v$, so any given node can only have neighbors once along a meta-path; elsewhere the update reduces to (9). And for a node having neighbors at a specific step along a meta-path, the corresponding node embedding prior to seeing neighbors will be zero by virtue of the initialization scheme. In this way, when neighbors do occur, (8) prevails. Hence the final embeddings produced at each node associated with rows of $\boldsymbol{X}$ will be compositions of (8) and (9). Given that a composition of positively homogeneous functions is also positively homogeneous, all weights can be pulled out in front without loss of generality. ∎

**Proposition G.2.** *Assume $\pi : \mathbb{R}^\kappa \to \mathbb{R}$ is affine, with weights in general position. Then for any $n > \kappa$, there exists a fixed ICL decoder $f_\theta$ such that*

$$\left| y_{test} - f_\theta \left( \{ \boldsymbol{x}_{i:}, y_i \}_{i=1}^n, \boldsymbol{x}_{test} \right) \right| = 0 \tag{10}$$

*with probability one over datasets $\mathcal{D}(\pi)$ generated according to Definition 4.1. Meanwhile, for any $r < d$, and any possible fixed encoder-decoder pair $\{g_\phi, f_\theta'\}$, we have*

$$\left| y_{test} - f_\theta' \left( \{ \boldsymbol{z}_{i:}, y_i \}_{i=1}^n, \boldsymbol{z}_{test} \right) \right| > 0, \quad \boldsymbol{z} = g_\phi(\boldsymbol{x}) \in \mathbb{R}^r \tag{11}$$

*with probability at least $1 - \binom{r}{k} / \binom{d}{k}$ over the same data generative distribution.*

*Proof:* We split the proof into two parts.

**Establishing (10).** By assumption, the stacked ICL samples are generated as $\boldsymbol{y} = \boldsymbol{X} \boldsymbol{w} + b = \widetilde{\boldsymbol{X}} \widetilde{\boldsymbol{w}} + b$, where $\widetilde{\boldsymbol{w}} \in \mathbb{R}^\kappa$ and $b \in \mathbb{R}$ are affine model parameters associated with $\pi$. Meanwhile $\boldsymbol{w}$ is simply $\widetilde{\boldsymbol{w}}$ padded with $d - \kappa$ zeros. From here, we first assume the $\kappa < n \leq d$. Let $\bar{\boldsymbol{X}}_n$ denote any set of $n$ columns selected from $\boldsymbol{X}$. Per the sampling process used to generate $\boldsymbol{X}$, it follows that $\text{rank}[\bar{\boldsymbol{X}}_n] = n$ almost surely for any such $\bar{\boldsymbol{X}}_n$; for reference, this is equivalent to the condition $\text{spark}[\boldsymbol{X}] = n + 1$ almost surely (Donoho & Elad, 2003). Note that if it were that $\text{rank}[\bar{\boldsymbol{X}}_n] < n$ with non-negligible probability, then it must be that $p(\boldsymbol{X})$ has unbounded density on a subspace within $\mathbb{R}^d$, which is disallowed by construction.

Then, by extension of Lemma 2 from Wipf & Rao (2004), it follows that $\boldsymbol{y} - b = \boldsymbol{X} \boldsymbol{w}$ is the unique equality involving a sparse weight vector satisfying $\|\boldsymbol{w}\|_0 < n$. Hence the decoder can check each combination of $n$ columns of $\boldsymbol{X}$ extracted from ICL samples to see if a sparse $\boldsymbol{w}$ vector allows for reconstructing observable labels $\boldsymbol{y}$ (note that there are more efficient ways to obtain $\boldsymbol{w}$ with additional assumptions; however, this is not necessary here for the proof). As any vector $\boldsymbol{w}$ so-obtained is unique and aligned with the ground-truth process, the model can then predict $y_{\text{test}}$ label at any test point $\boldsymbol{x}_{\text{test}}$.

Lastly, if $n \geq d$, then $\text{rank}[\boldsymbol{X}] = d$. Hence the ground-truth generative weights satisfy $\boldsymbol{w} = \boldsymbol{X}^\dagger \boldsymbol{y}$ such that again, perfect estimation of new test points is possible as before.

**Establishing (11).** The distribution of $\boldsymbol{X}$ is the same for each generated dataset (only the distribution of $\boldsymbol{y}$ changes by design). With an $r$-dimensional output, $q_\theta$ can reconstruct at most $r$-dimensions of $\boldsymbol{X}$. Per the assumed generative process, the proportion of datasets whereby all $\kappa$ columns fall within any group of $r$ fixed columns of $\boldsymbol{X}$ is $\binom{r}{k}/\binom{d}{k}$. Hence the encoder can achieve zero estimation error recovering in $\binom{r}{k}/\binom{d}{k}$ proportion of cases by $y_{\text{test}}$ by reconstructing any set of $r$ columns of $\boldsymbol{X}$, leaving a failure ratio of $1 - \binom{r}{k}/\binom{d}{k}$. But can we do any better than this?

Note that it is not possible to perfectly reconstruct any one coordinate of a given $\boldsymbol{x}$ from the others, i.e., there does not exist a function $\psi : \mathbb{R}^{d-1} \to \mathbb{R}$ such that $x_j = \psi(\boldsymbol{x}_{\backslash j})$ almost surely, where $\boldsymbol{x}_{\backslash j}$ denotes the elements of $\boldsymbol{x}$ excluding the $j$-th coordinate. (If such a function existed, then the density $p(\boldsymbol{x})$ would be unbounded.) Consequently, to achieve zero estimation error at test points, the encoder must be capable of reconstructing all dimensions of $\boldsymbol{x}$ associated with nonzero elements in $\boldsymbol{w}$. And since the encoder is required to stay fixed for all datasets, at best it can reconstruct $r$ columns, and so the success ratio from above cannot be improved upon. $\blacksquare$

**Proposition G.3.** *Let $\mathcal{F}$ denote the set of Lipschitz continuous functions on $[0,1]^\kappa$. Then there exists an ICL decoder $f_\theta$ such that (excluding smaller-order terms) we have*

$$\sup_{\pi \in \mathcal{F}} \mathbb{E}_{\mathcal{D}(\pi) \sim p}\left[y_{\text{test}} - f_\theta\big(\{\boldsymbol{x}_{i:}, y_i\}_{i=1}^n, \boldsymbol{x}_{\text{test}}\big)\right]^2 = O\left(n^{-2/\kappa}\right),\tag{12}$$

*where the data distribution $p$ is given by Definition 4.1 with $\mathcal{X} = [0,1]$. In contrast, with $\kappa = 1$ and $y = \widetilde{x}$, we have*

$$\inf_{f_\theta} \sup_{g_\phi \in \mathcal{B}} \mathbb{E}_{\mathcal{D}(\pi) \sim p}\left[y_{\text{test}} - f_\theta\big(\{\boldsymbol{z}_{i:}, y_i\}_{i=1}^n, \boldsymbol{z}_{\text{test}}\big)\right]^2 = \Theta\left(n^{-2/d}\right),\tag{13}$$

*where $\boldsymbol{z} = g_\phi(\boldsymbol{x})$ represents an encoder selected from the set of bi-Lipschitz bijections $\mathcal{B}$ mapping $[0,1]^d \to [0,1]^d$.*

*Proof:* The proof is segmented into two parts as follows.

**Establishing (12).** This bound follows by piecing together well-known results from learning theory. We begin by splitting into equal halves a given set of training samples $\{\boldsymbol{x}_{i:}, y_i\}_{i=1}^n$ from some $\mathcal{D}(\pi)$ with $S$ fixed. The high-level strategy is to define a set of nearest-neighbor estimators with the first half, and then select from among these estimators by enlisting the second half. Existing sample complexity results can then be applied to produce the final result.

To this end, for each subset of $\kappa$ elements from $d$ total input feature dimensions, we define the 1-nearest-neighbor estimator

$$f_{S'}(\boldsymbol{x}) = y_{i^*(\boldsymbol{x}, S')}, \quad \text{with } i^*(\boldsymbol{x}, S') = \arg\min_{i < \lceil n/2 \rceil} \left\|\boldsymbol{x}_{S'} - (\boldsymbol{x}_{i:})_{S'}\right\|\tag{14}$$

and subscript $S'$ denoting that only elements of any $\boldsymbol{x}$ within this subset are included. We then form a selection function $f_\theta(\boldsymbol{x}) = f_{S^*}(\boldsymbol{x})$ where

$$S^* = \arg\min_{S' \in \binom{[d]}{\kappa}} \left[\sum_{i=\lceil n/2 \rceil}^n \left(y_i - f_{S'}(\boldsymbol{x}_{i:})\right)^2\right].\tag{15}$$

Regardless of how each constituent $f_{S'}$ was originally constructed, the subsequent selector based on (15) follows the oracle inequality

$$\mathbb{E}\left[\left(y_{\text{test}} - f_{S^*}(\boldsymbol{x}_{\text{test}})\right)^2 \Big| S, \{\boldsymbol{x}_{i:}, y_i\}_{i<\lceil n/2 \rceil}\right] \leq \min_{S'} \mathbb{E}\left[\left(y_{\text{test}} - f_{S'}(\boldsymbol{x}_{\text{test}})\right)^2 \Big| S, \{\boldsymbol{x}_{i:}, y_i\}_{i<\lceil n/2 \rceil}\right] + O\left(\sqrt{\frac{\log\binom{d}{\kappa}}{n}}\right)$$

$$\leq \mathbb{E}\left[\left(y_{\text{test}} - f_S(\boldsymbol{x}_{\text{test}})\right)^2 \Big| S, \{\boldsymbol{x}_{i:}, y_i\}_{i<\lceil n/2 \rceil}\right] + O\left(\sqrt{\frac{\log\binom{d}{\kappa}}{n}}\right),\tag{16}$$

where the expectation is over $\{\boldsymbol{x}_{\text{test}}, y_{\text{test}}\}$ and $\{\boldsymbol{x}_{i:}, y_i\}_{i=\lceil n/2 \rceil}^n$ following dataset generative process from Definition 4.1 but for now with $S$ fixed. (i.e., the samples used in (14) are also treated as fixed here). See for example Section 5.4 of Hajek &

Raginsky (2021) for that standard and Hoeffding and union bounding process used to establish the first inequality in (16); the second inequality trivially follows from removing the min operator.

We next need to bound the r.h.s. expectation within (16). Adopting $i^*$ as shorthand for $i^*(\boldsymbol{x}_\text{test}, S)$, we have

$$\left| y_\text{test} - f_S(\boldsymbol{x}_\text{test}) \right| = \left| y_\text{test} - y_{i^*} \right| \leq L \left\| (\boldsymbol{x}_\text{test})_S - (\boldsymbol{x}_{i^*})_S \right\|, \tag{17}$$

which follows from the assumed Lipschitz continuity of $\pi$, recalling that $y = \pi(\boldsymbol{x}_S)$ by definition. We next square both sides and take an expectation over the samples $\{\boldsymbol{x}_{i:}, y_i\}_{i < \lceil n/2 \rceil}$ used in (14) as well as $\boldsymbol{x}_\text{test}$. These operations lead to

$$\mathbb{E}\left[ \left( y_\text{test} - f_S(\boldsymbol{x}_\text{test}) \right)^2 \big| S \right] \leq L^2 \mathbb{E}\left[ \left\| (\boldsymbol{x}_\text{test})_S - (\boldsymbol{x}_{i^*})_S \right\|^2 \big| S \right] \leq O\left( n^{-2/\kappa} \right), \tag{18}$$

where the right-most inequality stems from standard properties of nearest neighbors (e.g., see Theorem 2.1 in Biau & Devroye (2015)). Hence we can insert (18) into (16) after taking the expectation of both sides of the former w.r.t. $\{\boldsymbol{x}_{i:}, y_i\}_{i < \lceil n/2 \rceil}$. And lastly, because the resulting bound also holds for any given $S$, it also hold for all $S \in \binom{[d]}{\kappa}$. Hence we arrive at

$$\mathbb{E}\left[ \left( y_\text{test} - f_{S^*}(\boldsymbol{x}_\text{test}) \right)^2 \right] \leq O\left( n^{-2/\kappa} \right) + O\left( \sqrt{\frac{\log \binom{d}{\kappa}}{n}} \right), \tag{19}$$

with expectation over the entire $\mathcal{D}(\pi)$ generative process. This expression is dominated by the first term for $\kappa > 4$ as $n$ becomes large, completing the proof.

**Establishing (13).** Per the specified setup, we have $y = \widetilde{x} = [g_\phi^{-1}(\boldsymbol{z})]_i$ for some coordinate $i$, where $\boldsymbol{z} = g_\phi(\boldsymbol{x})$. For now we will assume that $i = 1$. Because $g_\phi$ is assumed to be a bi-Lipschitz, $g_\phi^{-1}$ is also Lipschitz over all of its coordinates, including the first. Therefore $h(\boldsymbol{z}) = [g_\phi^{-1}(\boldsymbol{z})]_1$ is a Lipschitz continuous function over domain $[0, 1]^d$. Additionally, because $p(\boldsymbol{x}) \geq p_\text{min}$, if follows that $p(\boldsymbol{z}) \geq p_\text{min}/L^d$, where $L$ is the upper Lipschitz constant associated with $g$. From here, it has already been established that the minimax squared estimation error for such an $h$, with input distribution bounded away from zero almost surely on a compact domain, is $\Theta\left( n^{-2/d} \right)$. Additionally, searching for an unknown $i$ incurs a modest additional cost such that the overall rate is the same. ∎

