# OpenReview forum: "No Need to Train Your RDB Foundation Model"
_ICML.cc/2026/Conference — ICML 2026 regular_

### Official Review · Reviewer_jozg · 2026-03-12

**Soundness:** 3
**Presentation:** 4
**Significance:** 3
**Originality:** 3
**Overall Recommendation:** 5
**Confidence:** 3

**Summary:**

The paper discusses the problem of generalizing predictive models to multi-table relational databases without retraining. The classical dense encoders employed in supervised learning are insufficient for ICL-based foundation models, as they mix informative and uninformative features before they have the label information provided. The authors tackle this by designing JUICE (Just Use Intra-Column Encodings) and the RDBLearn toolkit, which use SQL primitives for scalable, parameter-free relational featurization.

**Compliance With Llm Reviewing Policy:**

Affirmed.

**Final Justification:**

The authors satisfactorily answered my clarifying questions and increased my confidence in interpreting the experimental results.

Based on other reviews and the rebuttal discussion, the paper appears interesting and technically sound.

**Key Questions For Authors:**

1. Please clarify if 10,000-sample downsampling was applied to ICL-based models alone (RDBLearn and KumoRFM) or if the fully-supervised baselines (RelGT and 4DBInfer GNNs) were also limited to 10k samples? How did it impact the performance gap?

2. Since RDBLearn does not support text features, please clarify if LLM-based baselines (RelLLM, LLM-A/B) were evaluated using a "text-disabled" version of the datasets themselves, or did they keep textual attributes which RDBLearn omitted?

**Limitations:**

yes

**Strengths And Weaknesses:**

1. The paper is well-written and mostly easy to follow.
2. The authors address an important real-world problem of going beyond single-table foundation models to enterprise RDBs with complex data structures.
3. They build on well-supported statements (Propositions 3.2, 4.2, 4.3) that parameter-free encoders do not reduce the expressiveness in ICL, while dense encoders also increase sample complexity in out-of-distribution regimes.
4. RDBLearn, they propose, is competitive to fully supervised state-of-the-art (RelGT) models and outpaces existing foundation models on RelBench and 4DBInfer.

Still, some details of their experimental procedure remain unclear to me (see questions).

---

> ### Author Rebuttal · Authors · 2026-03-30
>
> Thanks for pointing out the significance, clarity, and solid foundation of our work.  The reviewer's only unresolved concerns distill into the following questions, which we address in turn.
>
> **Comment:**
> *Please clarify if 10,000-sample downsampling was applied to ICL-based models alone (RDBLearn and KumoRFM) or if the fully-supervised baselines (RelGT and 4DBInfer GNNs) were also limited to 10k samples? How did it impact the performance gap?*
>
> **Response:**
> Other methods were *not* limited to 10k samples, having access to the full training splits.  The only possible exception is KumoRFM, which as a closed-source model is characterized by some unknown implementation details.
>
> **Comment:**
> *Since RDBLearn does not support text features, please clarify if LLM-based baselines (RelLLM, LLM-A/B) were evaluated using a "text-disabled" version of the datasets themselves, or did they keep textual attributes which RDBLearn omitted?*
>
> **Response:**
> Great question to clarify.  Critically, all other approaches *had full access to textual features*, giving them a significant advantage, and yet RDBLearn is still able to outperform them in many cases.  For future work, equipping RDBLearn with analogous textual features provides an avenue for additional improvement.

---

> > ### Author Rebuttal · Reviewer_jozg · 2026-03-31
> >
> > I thank the authors for the clarifications.

---

> > > ### Author Response · Authors · 2026-04-08
> > >
> > > In the rebuttal acknowledgement, the reviewer selected option "(a) Fully resolved - My concerns have been adequately addressed."  We sincerely appreciate this positive assessment and support.  However, we also just noticed that the official review has been updated to say that some meaningful limitations remain with our work that prevent raising the score (see ``Final justification'' comment).  Given that the reviewer did not list any limitations, this statement is presumably related to our discussion with reviewer 8UoC regarding limited baseline testing on RelBench-v2 data.
> > >
> > > In this regard, we have recently provided new results to resolve this issue (see our latest response to reviewer 8UoC).  Notably, our RDBLearn framework achieves the best overall performance (relative to *both* supervised and foundation model approaches) on completely new RelBench-v2 tasks out of the box with no modifications or tuning whatsoever.  Moreover, we are also the first to test any RDB foundation models on RelBench-v2 tasks (most prior RDB FM work only considers RelBench-v1 tasks, whereas we now include tasks from RelBench-v1, 4DBInfer, and RelBench-v2).  In light of this new context, we respectively then ask if the final score for our paper might be reconsidered.  Thanks again for your continued engagement with our paper.

---

### Official Review · Reviewer_8UoC · 2026-03-13

**Soundness:** 3
**Presentation:** 1
**Significance:** 3
**Originality:** 3
**Overall Recommendation:** 4
**Confidence:** 5

**Summary:**

The paper introduces RDBLearn, a zero-shot foundation model for RDBs built on a parameter-free feature extractor called JUICE. The core idea is that simple 'vertical-only' compression preserves column identity better than complex GNN encoders, which often conflate features. By pairing these extracted features with a tabular decoder like TabPFN, the model achieves state-of-the-art results on RelBench and 4DBInfer. The work effectively shows that for zero-shot relational tasks, a minimal encoder can outperform heavy, pre-trained foundation models.

**Compliance With Llm Reviewing Policy:**

Affirmed.

**Key Questions For Authors:**

\[Q1\]  Could the authors provide experiments comparing JUICE against other ‘naive’ relational feature extraction methods to demonstrate that the specific ‘vertical-only’ design of JUICE contributes unique predictive value beyond simply providing a compatible input format for the downstream ICL decoder?
\[Q2\] Could the authors test their method on the SALT benchmark \[1, 2\] ? Since it is part of RelBench, it should be easy to run. I think it is a much more realistic setting as it was designed with industrial use cases in mind.

\[1\] Klein, T., Biehl, C., Costa, M., Sres, A., Kolk, J., & Hoffart, J. (2025). SALT: Sales autocompletion linked business tables dataset. arXiv preprint arXiv:2501.03413.
\[2\] Lachi, D., Mohammadi, M., Meyer, J., Arora, V., Palczewski, T., & Dyer, E. L. Integrating Temporal and Structural Context in Graph Transformers for Relational Deep Learning. In The Fourth Learning on Graphs Conference.

**Limitations:**

Yes

**Strengths And Weaknesses:**

Strengths:

* The paper provides a significant insight by showing that simple, parameter-free feature extraction can outperform complex, trained relational models in a zero-shot setting.
* The paper has extensive evaluation across diverse and large-scale benchmarks, including RelBench and 4DBInfer. The results are very compelling because they remain competitive with state-of-the-art supervised models (like RelGT).
* The method is very efficient. The paper focuses a lot on practical scalability, showing that high-quality features can be extracted using simple SQL primitives without the need for expensive training.

Weakness:

* The paper looks really unpolished. All tables are included as low-quality screenshots rather than formatted text. This is unexpected for a top-tier conference submission and significantly hinders readability.
* The method’s novelty is primarily focused on a feature-engineering pipeline (JUICE) that restructures relational data, effectively delegating the 'intelligence' to existing tabular models like TabPFN. I do not see this as a major weakness, and it does not need to be addressed beyond the experiment I have requested in Q1.

---

> ### Author Rebuttal · Authors · 2026-03-30
>
> We appreciate the reviewer pointing out the significant insights and evaluations in our work.  We address critiques/questions below.
>
> **Comment:**
> *The paper looks really unpolished. All tables are low-quality screenshots ...*
>
> **Response:**
> We can readily update the tables to higher-resolution formatted text.
>
> **Comment:**
> *[Q1] Could the authors provide experiments comparing JUICE against other ‘naive’ relational feature extraction methods ...*
>
> **Response:**
> This is a wonderful suggestion that serves to solidify the foundational premises of RDBLearn.  Along these lines, to instantiate a relational feature extractor that explicitly *violates* the vertical-only design of JUICE, we adopted the randomly-initialized GNN architecture included with the RelBench-v2 release (the model also includes a pre-trained encoder for text fields as well).  This selection is motivated by substantial literature advocating for the efficacy of randomized-GNN layers specifically as viable feature encoders for prediction tasks (e.g., [Bui et al., 2025](https://arxiv.org/html/2502.00190v1); [Degraeve et al., 2022](https://arxiv.org/abs/2203.02424); [Xu et al., 2023](https://proceedings.mlr.press/v202/xu23w/xu23w.pdf)). In the present context, such randomized GNN filters explicitly *mix cross-column tabular information in direct contrast to JUICE*.  Multi-table features so-obtained are then passed to TabPFN-v2.5 for making final predictions.  We compare this new baseline against JUICE features applied to an identical TabPFN-v2.5 prediction head for standardized comparison.
>
> Results are shown in the table below across the RelBench tasks from our submission.  Two things are evident from these results that help confirm our original analysis.  First, Rand-GNN performs quite well (e.g., relative to RDB FMs in prior work), consistent with our analysis-based expectation that trained encoders may be less important in ICL settings (as opposed to supervised settings).  And second, *cross-column mixing is consistently unhelpful*, hence JUICE is uniformly better than Rand-GNN.  We will definitely add these results to a revision to strengthen the paper.
>
> | Dataset | Task | Rand-GNN  |JUICE |
> |---|---|---|---|
> | | **Classification (AUC)** | | |
> | rel-amazon | item-churn |77.73 | **81.73** |
> | | user-churn |65.87 |**68.23** |
> | rel-avito | user-clicks |61.58 | **67.1** |
> | | user-visits |64.15 |**65.55** |
> | rel-hm | user-churn |66.54 |**67.79** |
> | rel-stack | user-badge |83.14 |**84.36** |
> | | user-engage |85.14 |**89.29** |
> | rel-trial | study-outcome |55.98 |**71.64** |
> | | **Regression (MAE)** | | |
> | rel-amazon | item-ltv |66.13 |**48.50** |
> | | user-ltv |17.58 |**14.52** |
> | rel-avito | ad-ctr |0.038 |**0.034** |
> | rel-hm | item-sales |0.065 |**0.063** |
> | rel-stack | post-votes |0.071 |**0.068** |
> | rel-trial | site-success |0.434 |**0.406** |
> | | study-adverse |53.37 | **44.51** |
>
> **Comment:**
> *[Q2] Could the authors test their method on the SALT benchmark [1, 2] ? Since it is part of RelBench, it should be easy to run ...*
>
> **Response**
> As the reviewer points out, certain predictive tasks have been extracted from the SALT dataset and codified within RelBench; however, this only occurred in [RelBench-v2](https://arxiv.org/abs/2602.12606) released on 2026/02/13.  Moreover, unfortunately to date none of the other existing RDB foundation models we are aware of have provided implementations or performance results using these new data to which we can compare against.
>
> Still, we sympathize with the reviewer's larger point that additional benchmarking can increase practical relevance.  Of the 4 new datasets added to RelBench-v2, we conduct new experiments using all tasks from the rel-RateBeer benchmark.  We chose this dataset because of implementational barriers to adopting the remaining 3 (rel-SALT involves multi-class prediction tasks with greater than 10 classes, which our current pipeline does not support; rel-Mimic requires a complex authentication process to even access the data; rel-arXiv is text based and hence better suited for LLM approaches).  These issues can be resolved in the future, but during the short rebuttal period we focus on the more accessible tasks.
>
> We run our exact same RDBLearn pipeline, *with zero tuning or modifications of any kind*, on all rel-RateBeer tasks, with results shown below.  Baselines LightGMB and GNN results come from the [RelBench-v2 paper](https://arxiv.org/abs/2602.12606).  Even with full per-dataset supervision for these baselines, RDBLearn performance *with no training* is still better in most cases.
>
> | Task | LightGBM (supervised) | GNN (supervised) | RDBLearn |
> |---|---|---|---|
> |**Classification (AUC)**| | | |
> | user_churn |83.92| 94.27 | **97.81** |
> | beer_churn |76.21| 78.67 | **84.92** |
> | brewer-dormant | 75.79 | 80.51 | **81.68** |
> |**Regression (MAE)**| | | |
> | user-count | 20.35 | 7.374 | **7.231** |
> | beer_ratings-total_score | 0.447 | **0.323** | 0.434 |

---

> > ### Author Rebuttal · Reviewer_8UoC · 2026-04-01
> >
> > Thank you for the clarification. While I understand that other foundation models have not yet reported results on this benchmark, I am not fully convinced that the recency of the benchmark alone prevents evaluation. Even if other foundation models have not been tested yet, the model could still be compared against standard supervised baselines on these tasks.
> >
> > Additionally, it should be possible to take an existing checkpoint (e.g., from Griffin) and evaluate it directly on these datasets to provide a point of comparison.
> >
> > Since I have already given a weak accept, I will keep my current score unless this concern is addressed.

---

> > > ### Author Response · Authors · 2026-04-05
> > >
> > > Thanks for the additional feedback; we address the follow-up reviewer comments as follows.
> > >
> > > **Comment:**
> > > *Thank you for the clarification. While I understand that other foundation models have not yet reported results on this benchmark, I am not fully convinced that the recency of the benchmark alone prevents evaluation.*
> > >
> > > **Response:**
> > > We completely agree that recency does not prevent evaluation.  Our point is just that this new standardized benchmark was not available at the time of our submission (being posted to arXiv in February after the ICML deadline), and therefore would not traditionally be viewed as a necessary comparison to consider.  Even so, its inclusion does enhance our work as we concede below.
> > >
> > > **Comment:**
> > > *Even if other foundation models have not been tested yet, the model could still be compared against standard supervised baselines on these tasks. Additionally, it should be possible to take an existing checkpoint (e.g., from Griffin) and evaluate it directly on these datasets to provide a point of comparison.*
> > >
> > > **Response:**
> > > We now compare RDBLearn on all RelBench-v2 tasks from the rel-RateBeer dataset against *four* baseline models. These include the two supervised models from the official RelBench-v2 release, namely, LightGBM and a heterogeneous GNN. Meanwhile, since no prior RDB foundation model results exist on RelBench-v2, we adapted and executed Griffin (per the reviewer's suggestion) and the new Rand-GNN baseline (see rebuttal) ourselves.  For Griffin, we pre-processed all datasets using multiple scripts as required by the official Griffin repo for format conversion.  We then tested all available Griffin checkpoints and selected the best performing model.  We present rel-RateBeer comparative results in the table below, and specifically address the performance of each RDB foundation model as follows:
> > >
> > > * *Griffin*: From the table, Griffin accuracy is lowest on brewer-dormant, a task quite distinct from the original RelBench-v1 tasks on which Griffin pre-training is based. On the two churn-related tasks Griffin performance is a bit better though, likely due to greater overlap with pre-training, but still failing to match the other supervised models; again this is expected as no Griffin pre-training or fine-tuning has thus far been conducted using RelBench-v2.   Of course if fine-tuning were introduced within the Griffin pipeline (as generally advocated in Wang et al., 2025), the performance would likely improve considerably.  But this possibility remains well outside of our current scope of *supervision-free* RDB foundation models.
> > >
> > > * *Rand-GNN*:  Performance is relatively strong compared to Griffin, even competitive with supervised models in some cases.  This is not surprising given that the average AUC performance of Rand-GNN on the original RelBench-v1 tasks (70.02 mean AUC; see our rebuttal for individual AUC values) is actually higher than many previously-published RDB foundation models we report in Figure 2 of our submission (in fact, from Figure 2 we observe that only KumoRFM at 72.72 mean AUC is higher than Rand-GNN).
> > >
> > > * *RDBLearn*:  On these new RelBench-v2 tasks, RDBLearn has the best overall performance directly out of the box, even rivaling the supervised approaches.  Again, we attribute this to frontier tabular ICL combined with theoretically-motivated RDB featurization that is not tied to any domain-specific scenarios (as in Griffin pre-training) nor cross-column mixing (as with Rand-GNN).
> > >
> > >
> > > | Task | LightGBM (supervised) | GNN (supervised) |Griffin (foundation) |Rand-GNN (foundation) | RDBLearn (foundation)|
> > > |---|---|---|---|---|---|
> > > |**Classification (AUC$\uparrow$)**| | | | |
> > > | user-churn |83.92| 94.27 | 69.38 |89.73| **97.81** |
> > > | beer-churn |76.21| 78.67 | 57.38 |76.87| **84.92** |
> > > | brewer-dormant | 75.79 | 80.51 |39.30 |78.75| **81.68** |
> > > |**Regression (MAE $\downarrow$)**| | | | |
> > > | user-count | 20.35 | 7.374 |13.01|13.94 | **7.231** |
> > > | beer_ratings-total_score | 0.447 | **0.323** |0.739 |0.466| 0.434 |
> > >
> > > We close by reiterating that the RelBench-v2 release (at the time of this writing) only includes two baseline models (LightGBM and GNN-based), and as of yet no follow-up work has provided additional models according to google scholar, foundation or otherwise.  Hence by conducting comparisons here with three new models (Griffin, Rand-GNN, RDBLearn) we are taking useful steps to further RDB benchmarking.
> > >
> > >
> > > **Comment:**
> > > *Since I have already given a weak accept, I will keep my current score unless this concern is addressed.*
> > >
> > > **Response:**
> > > As this is the last chance for us to respond per ICML guidelines, we hope that these new results have adequately addressed the reviewer's comments.  Thanks again for the continued engagement and constructive pointers.  We may respectfully quibble a bit about RelBench-v2 release timing, but regardless, our paper has certainly been improved by taking it into consideration here as the reviewer suggested.

---

### Official Review · Reviewer_FtLU · 2026-03-13

**Soundness:** 3
**Presentation:** 3
**Significance:** 3
**Originality:** 3
**Overall Recommendation:** 4
**Confidence:** 4

**Summary:**

This paper introduces a new approach to building foundation models for relational databases (RDBs), challenging the conventional wisdom that complex, trainable encoders (like GNNs) are necessary for in-context learning (ICL).

The core contribution is the JUICE encoder, which operates on a simple principle: compress vertically within columns, but never mix information horizontally across columns. By applying fixed aggregation functions (e.g., sum, mean) to individual data columns along predefined meta-paths, JUICE produces interpretable, column-aligned embeddings without any trainable parameters. This design preserves the identity of each column and defers the task of identifying relevant features to the downstream model.

The authors implement this idea in an open-source toolkit called RDBLearn, which pairs the JUICE encoder with a powerful single-table ICL model (like TabPFN). This combination works out-of-the-box on new databases without any fine-tuning.

The paper provides theoretical proofs showing that fixed, dense encoders that mix columns inevitably introduce errors in ICL settings. Extensive experiments on benchmarks like RelBench and 4DBInfer demonstrate that this parameter-free approach not only outperforms existing closed-source foundation models but also rivals fully-supervised methods, all while requiring zero training on the target database.

**Compliance With Llm Reviewing Policy:**

Affirmed.

**Key Questions For Authors:**

Q1. A core limitation of JUICE's strict per-column isolation is its inability to perform conditional aggregations that require cross-column logic before summarization. For example, computing the "Sum of Amount WHERE Status='Active'" requires filtering by the 'Status' column while aggregating the 'Amount' column. When such signals are critical, does the downstream TabPFN decoder have the capacity to reliably infer these conditional relationships from the separately aggregated statistics (e.g., from separate "sum(Amount)" and "count(Status='Active')" features)?

Q2. JUICE's design of computing multiple aggregations (A) for every meta-path (M) and column (K) leads to an output feature dimension that scales as O(K × M × A). For enterprise-scale RDBs with dense schemas (e.g., 50+ tables, numerous relationships), this could potentially exceed the practical context length limits of Transformer-based ICL models like TabPFN. How does RDBLearn address this scalability challenge in practice? Are there built-in feature selection mechanisms, pruning strategies, or dimensionality reduction techniques employed to ensure the method remains applicable to large-scale, real-world databases? If not, what is the estimated upper bound on schema complexity that RDBLearn can handle with current hardware constraints? Addressing this point would clarify the method's real-world applicability.

**Limitations:**

yes

**Strengths And Weaknesses:**

Strengths:

S1. The paper presents a clear and well-defined theoretical framework for RDB encoding under the ICL setting. The JUICE encoder is rigorously motivated by the principle that horizontal compression across columns should be avoided before label information is available. Theoretical propositions (e.g., Proposition 3.2 on reparameterization without weights, Proposition 4.2 and 4.3 on the limitations of dense encoders) are well-articulated and supported by reasonable assumptions. The proofs in Appendix G are logically structured and appear correct.

S2. The RDBLearn framework is a practical contribution that could be widely adopted by practitioners, especially given its ease of use and strong performance out of the box.

S3. The experimental design is comprehensive: the paper evaluates on multiple benchmarks (RelBench, 4DBInfer), compares against a wide range of baselines (supervised, ICL-based, LLM-based), and includes ablation studies (e.g., varying hops, base models, aggregation functions) to support claims.

Weaknesses:

W1. While Propositions 4.2 and 4.3 provide valuable lower bounds on the risks of fixed feature mixing, their assumptions model the encoder as a non-adaptive transformation. This may be a simplification, as modern relational foundation models often leverage self-supervision to learn semantic compression that could potentially discriminate between informative and uninformative features even in OOD settings. Addressing this gap between the theoretical model and practical representation learning would enhance the paper's impact.

W2. By design, JUICE strictly prohibits any cross-column interaction during the encoding phase. While this preserves column identity and prevents harmful mixing, it also imposes a limitation: the encoder cannot perform conditional aggregations that require joint consideration of multiple columns. For example, computing the "Sum of Amount WHERE Status='Active'" is not expressible in JUICE's isolated, per-column processing paradigm. This constraint means that any logic requiring column interdependence prior to aggregation must be entirely inferred by the downstream TabPFN decoder from the separately aggregated statistics. In schema topologies where such conditional signals are critical, this architectural choice may limit performance. A discussion on how JUICE's design handles (or trades off) such logic-dependent patterns would strengthen the paper.

W3. While the parameter-free extraction is efficient, JUICE's design of preserving all column-wise information can lead to a rapid expansion of the feature space. For instance, with K columns, M meta-paths, and A aggregators, the output dimension scales as O(K × M × A). This characteristic shifts the burden of feature selection entirely onto the downstream TabPFN decoder. However, Transformer-based architectures like TabPFN have inherent limitations, including fixed maximum token lengths and quadratic complexity in the context window. For enterprise-scale relational databases with dense schemas (e.g., 50+ tables), the resulting feature dimension could potentially exceed these practical hardware limits, which may constrain the method's applicability in such large-scale scenarios. The paper would benefit from a discussion on how RDBLearn handles this scalability challenge, such as feature pruning strategies or dimensionality reduction techniques.

W4. While Proposition 3.2 shows that weights can be removed under positive homogeneity, the paper does not deeply explore whether certain aggregation functions might conflict or degrade performance in practice. The choice of aggregators (sum, mean, mode, etc.) is reasonable but somewhat heuristic.

W5. The time-aware encoding (relative temporal differences) is mentioned but not deeply analyzed. It would strengthen the paper to include a dedicated experiment or case study showing how this affects performance in temporal OOD scenarios.

W6. Some parts of the paper are dense and may be difficult for non-expert readers to follow, particularly the theoretical sections and the meta-path GNN formulation. A model architecture diagram or intuitive explanation of Proposition 3.2 and 4.2 in the main text would improve accessibility.

W7. The reliance on synthetic pre-training for the ICL decoder (e.g., TabPFN) raises questions about how well the approach would scale to tasks with very different feature distributions or complex text-heavy columns. This is acknowledged but not deeply explored.

---

> ### Author Rebuttal · Authors · 2026-03-30
>
> Thanks for the many constructive comments.
>
> **Comment:**
> *W1. While Propositions 4.2 and 4.3 provide valuable lower bounds ...*
>
> **Response:**
> Apologies if we misunderstood the reviewer's point here, and we are happy to provide further clarification as needed.  However, under our foundation model setting, the implicit assumption of any RDB FM encoder is that the weights are necessarily *fixed* (non-adaptive), meaning they do not change from dataset to dataset or task to task (i.e., irrespective of the loss used for pre-training). Here the relevance of any particular input feature remains indeterminate without label info, as the exact same set of input features can have a completely different relevance profile depending on downstream tasks/labels (e.g., as in Fig. 1).
>
> That being said, we concede that there may still exist cases in practice where the utility of certain RDB columns can be at least partially determined by a reasoning model with access to rich task and column descriptions (a loose proxy for label info).  Of course this sort of complementary reasoning could be exploited by RDBLearn as well to filter out likely uninformative columns, simplifying the ICL stage (a worthwhile pursuit for future versions of RDBLearn).
>
> **Comment:**
> *W2. By design, JUICE strictly prohibits any cross-column interaction ...*
>
> **Response:**
> The reviewer's point is well-taken, and we acknowledge that restriction to vertical compression does have the potential to exclude useful cross-column features at times.  Even so, any encoder compression will involve discarding information, and so if the encoder is fixed (as required for a foundation model) it will always be possible to define edge cases where the discarded information turns out to be useful.  Our central point then is merely that vertical compression within high dimensional columns (sharing units and roles) is a reasonable compromise absent dataset- and task-specific information that can be exploited by supervised training.
>
> **Comment:**
> *W3. While the parameter-free extraction is efficient ...*
>
> **Response:**
> While none of the current benchmarks push upper boundaries, it is true that tables with many columns and associated meta-paths could eventually pose computational issues (although TabPFN and related models are constantly being scaled upwards).  One practical workaround is to lean on ensembling, whereby sub-sampled feature columns are repeatedly passed to the RDBLearn framework in parallel and then combined for making final predictions.  Indeed, for single tables TabPFN has been shown to work well within an ensembling paradigm (Hollmann et al., 2025), and no training is required when constructing each constituent predictor.
>
> **Comment:**
> *W4. While Proposition 3.2 shows ...*
>
> **Response:**
> While we have admittedly not conducted a thorough exploration over diverse combinations of aggregation functions, the current choices follow from three-fold motivation: (i) Their simplicity leads to intuitive features and downstream interpretability; (ii) they cover quantities that may reasonably reflect some degree of both homophily and heterophily relationships (see Lines 231-247); and (iii) prior GNN work suggests that these types of aggregations can be useful in practice (Corso et al., 2020).  Regardless, it is still possible that alternative aggregation sets could further improve performance; definitely worth considering for future work.
>
> **Comment:**
> *W5. The time-aware encoding ...*
>
> **Response:**
> For most cases, temporal encoding type is a minor factor, with results using absolute vs differential encodings varying by 1% or less.  However, on 3 tasks across all experiments, the gap exceeded 5%, and on each of these cases differential encoding was superior (5.4% on Retailrocket-CVR, 5.7% on Rel-Hm/Item-Sales, and 12.8% on Rel-Avito/Ad-Ctr).  This indicates a modest degree of additional robustness, hence our selection of temporal difference encodings.  We can add these results to the revision.
>
>
> **Comment:**
> *W6. Some parts of the paper are dense ...*
>
> **Response:**
> Good suggestions; we can easily add such content to the revision.
>
> **Comment:**
> *W7. The reliance on synthetic pre-training ...*
>
> **Response:**
> We surmise that foundation models pre-trained using existing RDB benchmarks (as opposed to synthetic data alone) may be more sensitive to feature distribution shifts outside of these datasets.  Meanwhile, RDBLearn exhibits a degree of robustness here given that it works well on diverse real-world RDB datasets with no possibility of label leakage or covariate alignment by design.  That being said, we agree with the reviewer that for RDBs with text-heavy columns, and task labels dependent on such columns, RDBLearn in its present form would not be ideal.  Fortunately, RDBLearn can be adapted to include text encodings in future work.
>
> **Comment:**
> *Key Questions for Authors:  Q1 and Q2 ...*
>
> **Response:**
> Please see responses above which address these questions.

---

### Decision · Program_Chairs · 2026-04-30

**Decision:**

Accept (regular)

**Comment:**

The paper presents RDBLearn, a zero-shot foundation model for relational databases (RDBs), built upon a parameter-free feature extractor called JUICE. The key insight is that simple "vertical-only" compression better preserves column identities compared to complex GNN encoders, which often mix and conflate features. By combining these extracted features with a tabular decoder such as TabPFN, the model achieves state-of-the-art performance on tasks like RelBench and 4DBInfer. The study demonstrates that for zero-shot relational tasks, a lightweight, minimal encoder can outperform large, pre-trained foundation models.

This is a solid work.